



# Paleoclimate in continental northwestern Europe during the Eemian and Early-Weichselian (125-97 ka): insights from a Belgian speleothem.

Stef Vansteenberge[1], Sophie Verheyden[1,2], Hai Cheng[3,4], Lawrence R. Edwards[4], Eddy Keppens[1] and Philippe Claeys[1]

*Correspondence to:* S. Vansteenberge (svsteenb@vub.ac.be)

[1]Earth System Science Group, Analytical-, Environmental- & Geo-Chemistry, Vrije Universiteit Brussel, Brussels, Belgium
[2]Royal Belgian Institute for Natural Sciences, Brussels, Belgium
[3]Institute of Global Environmental Change, Xi'an Jiaotong University, Xi'an, China
[4]Department of Earth Sciences, University of Minnesota, Minneapolis, USA

**Abstract.** The Last Interglacial serves as an excellent time interval for studying climate dynamics during past warm periods. Speleothems have been successfully used for reconstructing the paleoclimate of Last Interglacial continental Europe. However, all previously investigated speleothems are restricted to southern Europe or the Alpine region, leaving large parts of northwestern Europe undocumented. To better understand regional climate changes over the past, a larger spatial coverage of European Last Interglacial speleothems is essential. Here, we present new, high-resolution data from a stalagmite (Han-9) obtained from the Han-sur-Lesse cave in Belgium. The Han-9 formed between 125.3 and ~97 ka, with interruptions of growth occurring at 117.3 – 112.9 ka and 106.6-103.6 ka. The speleothem was investigated for its growth, morphology and stable isotope ($\delta^{13}$C and $\delta^{18}$O) content. Speleothem formation within the Last Interglacial started relatively late in Belgium, as this is the oldest sample of that time period found so far, dated at 125.3 ka. Other European continental archives suggest that Eemian optimum conditions were already present during that time, therefore it appears that the initiation of the Han-9 growth is caused by an increase in moisture availability, linked to wetter conditions around 125.3 ka. The $\delta^{13}$C and $\delta^{18}$O proxies indicate a period of relatively stable conditions after 125.3 ka, however at 120 ka the speleothem $\delta^{18}$O registered the first signs of regionally changing climate conditions, being a modification of ocean source $\delta^{18}$O linked to an increase in ice volume towards the MIS 5e-5d transition. The end of the Eemian is marked by drastic vegetation changes recorded in the speleothem $\delta^{13}$C at 117.5 ka, immediately followed by a stop in speleothem growth at 117.3 ka, suggesting that climate became significantly dryer. The Han-9 record covering the Early-Weichselian displays larger amplitudes in both the isotope proxies and the stalagmite morphology, evidencing increased variability compared to the Eemian. Greenland Stadials are recognized in the Han-9 and the chronology is consistent with other European (speleothem) records. Greenland Stadial 25 is reflected as a cold/dry period within the stable isotope proxies and the second interruption in speleothem growth occurs simultaneously with Greenland Stadial 24.

**Keywords:** paleoclimate, Last Interglacial, Eemian, northwestern Europe, speleothem, stable isotopes, millennial variability, Greenland Stadials





## 1 Introduction

The Last Interglacial (LIG) period is known as the time interval before the Last Glacial period during which temperatures were similar or higher than present and the past Holocene period. In marine sediment cores, the LIG is defined as Marine Isotope Stage 5e (Shackleton, 1969). The start and end of the MIS5e period are conventionally set at 130 and 116 ka, respectively, based on marine records (e.g. Martinson et al., 1987). The expression of the LIG in continental western Europe is defined as the Eemian, although it does not coincide precisely with the isotopically constrained MIS 5e (Otvos, 2015). Given the ongoing debate about the Last Interglacial nomenclature, clarification about the terms used in this manuscript are required. This study focusses on speleothem archives, thus the terms "Eemian" and "Weichselian" are preferred in the context of European continental paleoclimate. The Eemian is defined as the optimum or acme LIG climate conditions (the "sensu stricto" definition). Subsequent to the Eemian, the Weichselian starts, with the Early-Weichselian in continental records corresponding to the time-equivalent of MIS 5d – 5a. "Glacial inception" is considered to be informal and only marks the Eemian to Early-Weichselian transition.

The Eemian was originally defined by Harting (1875) and was characterized by the occurrence of warm water mollusks in marine sediments of the Eem river valley, near Amsterdam, the Netherlands (Bosch et al., 2000). Nowadays, the Eemian is mostly interpreted as an interval of climate amelioration associated with the spread of temperate mixed forests in areas with similar vegetation today (Kukla et al., 2002). However, the Eemian is also known to be a diachronous unit (Kukla et al., 2002; Wohlfarth et al. 2013), with a longer duration up to 20 ka, from 130 to 110 ka, in southern Europe as evidenced by pollen records (Sanchez Goñi et al., 1999; Tzedakis et al., 2003). The LIG period exhibited global mean temperatures (GMT) 1.5° to 2° C higher than the pre-anthropogenic average together with peak eustatic sea levels that were between 5.5 and 9 m higher than present (Dutton and Lambeck, 2012). Therefore, the LIG gained a lot of attention from both paleoclimate and climate modelling communities for studying a warmer climate state and potential future sea-level rise (Loutre et al., 2014; Goelzer et al., 2015), even though configuration of Earth's orbital forcing parameters was different (Berger and Loutre, 2002). Subsequent to the Eemian, climate went into a glacial mode known as the Last Glacial cycle, or the Weichselian in the western European continental terminology, which lasted until the Holocene. A major control on climate variability during the Last Glacial is the occurrence of millennial scaled, rapid cold-warm-cold cycles, known as Dansgaard-Oeschger (D/O) events (Bond et al., 1993). These D/O cycles are expressed as a succession Greenland Stadial (GS) and Interstadial (GIS) phases in Greenland ice cores (Dansgaard et al., 1993; NGRIP members, 2004), as Atlantic Cold Events registered in sea-surface temperature proxies. Some of the Stadials are also associated with increases in the flux of Ice Rafted Debris (IRD) in the North-Atlantic ocean (McManus et al., 1994). These events have been linked to changes in the strength and shifts in the northwards extent of the Atlantic Meridional Overturning Circulation (AMOC) (Broecker et al., 1985). The exact cause of such changes in the AMOC mode are still debated. Nevertheless, according to Barker et al. (2015), it is more likely a non-linear response of a gradual cooling than a result of enhanced fresh-water input by iceberg calving, as previously proposed by Bond et al. (1995) and van Kreveld et al. (2000). Likewise, continental pollen records extracted from cores of Eifel Maar lakes (Sirocko et al., 2005) or peat bogs in the Vosges, France (Woillard, 1978; de Beaulieu and Reille, 1992; de Beaulieu, 2010), have recorded changes in pollen assembly, attributed to D/O variability. So far, up to 26 GS's have been identified in ice-cores, with GS 26 recognized as corresponding to the end of the Eemian Interglacial period (NGRIP members, 2004).



Speleothems have often been used for Late-Quaternary paleoclimate studies because of the ability of constructing accurate chronologies with U/Th dating up to 600 ka and their potential of holding high resolution, up to seasonal scale, paleoclimate signals (Fairchild and Baker, 2012). The speleothem records covering the Eemian and Early-Weichselian in Europe have provided detailed paleoclimate reconstructions (Genty et al.,

2013). These have shown that Eemian optimum conditions are indeed registered in European speleothem stable isotope proxies (Meyer et al., 2008; Couchoud et al., 2009) and that D/O climate events are also recorded during the Early-Weichselian (Bar-Matthews et al., 1999; Drysdale et al., 2007; Boch et al. 2011). Yet so far, all records covering that time period are located in southern Europe (Italy, Southern France, Levant) or the Alpine region. This study presents a new high-resolution speleothem dataset from Belgium in order to expand the European

coverage of Last Interglacial speleothem archives northwards.

Earlier chronostratigraphic work on speleothem deposits and detrital sediments within Belgian caves marked the presence of glacial/interglacial cycles, with speleothem formation restricted to interglacial periods, when warm and wet climates favored growth. Detrital sediments settle in colder periods, with river deposits in cold wet periods and reworked loams during cold dry periods (Quinif, 2006). From the 1980's onwards, speleothems

covering MIS 9 to 1 have been found in various Belgian caves (Bastin and Gewelt, 1986; Gewelt and Ek, 1988). Quinif and Bastin (1994) analyzed an Eemian flowstone from the Han-sur-Lesse cave for its pollen content, and have shown that vegetation above the cave area reflects interglacial climate optimum conditions around 130 +/- 10 ka. However, the dating of this material contains large uncertainties related to the alphaspectrometric methodology used. This study focusses on a recently obtained speleothem, the Han-9 from the Han-sur-Lesse

cave in Southern Belgium. This 70 cm long stalagmite was analyzed to better constrain 1) the chronology of Eemian optimum conditions in Belgium and 2) the occurrence and signature of millennial (D/O) scaled climate variability over northwestern Europe during the Early-Weichselian.

## 2  Han-sur-Lesse cave: geology and cave parameters

The Han-sur-Lesse cave system is the largest known subterranean karst network in Belgium, with a total length

of ~10 km. It is located within the *Calestienne*, a SW-NE trending superficial limestone belt of Middle Devonian age. After deposition, these Paleozoic sediments underwent Hercynian folding followed by Mesozoic erosion. The current hydrographic network was established during the Neogene and Pleistocene, by erosion into these folded belts (Quinif, 2006). The cave system was formed within the *Massif du Boine*, part of an anticline structure consisting out of Middle to Late Givetian reefal limestones, by a meander shortcut of the Lesse river

(Fig 1B). The thickness of the epikarst zone above the cave is estimated to be around 40 m.

The area of the Han-sur-Lesse cave is located ca. 200 km inland at an elevation of 200 m a.s.l. (Fig. 1) and is marked by a maritime climate with cool summers and mild winters. For the period 1999-2013, average year-temperature above the cave was 10.2 °C and average yearly rainfall amount 820 mm/year, which is spread over the entire year (Royal Meteorological Institute, RMI). The area above the cave mainly consists out of C3 type

vegetation with *Corylus*, *Fagus* and *Quercus* trees and as a natural reserve it has been protected from direct human influence for over 50 years (Timperman, 1989). The Lesse river enters the cave system at the *Gouffre de Belveaux* and exits at the *Trou de Han* approximately 24 hours later. The Han-Stm-9 stalagmite was collected within the *Réseau Rénversé*, the most distal part of the *Réseau Sud* or the southern network of the Han-Sur-Lesse cave system (Fig. 1B). The natural connection with other parts of the Han-sur-Lesse cave is fully submerged, but

in 1960 an artificial tunnel was established facilitating the accessibility (Timperman, 1989). When the Lesse river water is high, part of the stream is redirected through the southern network (Bonniver, 2010). The *Réseau*





*Renversé* does not contain a stream. Earlier studies have shown that cave drip waters are mostly supplied by diffuse flow through the host rock (Bonniver et al., 2011; Van Rampelbergh et al., 2014). The Han-sur-Lesse cave is partly accessible for tourists, but because of the difficult access, the southern network is protected from

any anthropogenic influence. Short term temperature logging in the southern network shows an average cave temperature of 9.45°C with a standard deviation < 0.02 °C, reflecting the average temperature of 9.2 °C above the cave for 2013. Minimum and maximum temperatures were 9.39 °C and 9.51 °C, respectively (C. Burlet, pers. comm.). In contrast, recent cave monitoring in more ventilated parts of the cave indicated a temperature seasonality of 3 °C (Van Rampelbergh et al., 2014).

The Han-sur-Lesse cave received scientific attention in the last decades, making it the best understood cave system in Belgium. This includes detailed hydrographic studies (Bonniver et al., 2010) and extended cave monitoring surveys (Verheyden et al., 2008; Van Rampelbergh et al., 2014), leading to successful paleoclimate reconstructions on Holocene speleothems down to seasonal scale (Verheyden et al., 2006; 2012; 2014; Van Rampelbergh et al., 2015). The elaborate cave monitoring has provided a steady base for understanding the cave

system and to interpret its paleoclimate records, even back to ~120 ka.

### 3 Methods and analytical procedures

All ages were acquired by using U/Th dating at the University of Minnesota Earth Sciences Department, Minneapolis. Nine samples were analyzed in 2013 and an additional batch of 14 samples was dated in 2015 to further improve the age-depth model and the time series, with locations chosen in function of the preliminary age

model and the stable isotope ($\delta^{13}$C and $\delta^{18}$O) data (Fig. 2D). For all U/Th analyses, 150-200 mg of speleothem calcite was milled and analyzed with a *Neptune* MC-ICP-MS from *Thermo-Scientific* at the University of Minnesota. The used half-lives for $^{230}$Th and $^{234}$U are reported in Cheng et al. (2013). Ages were corrected assuming an initial $^{230}$Th/$^{232}$Th ratio of 4.4 ±2.2 x $10^{-6}$. For additional information about the applied method, see Edwards et al. (1987) and Cheng et al. (2013) and references therein. Age-depth modeling was carried out using

the StalAge algorithm of Scholz and Hoffmann (2011). All depths are expressed in 'mm dft' with dft being 'distance from top'.

For stable isotope analysis, a total of 1118 samples were drilled with a *Merchantek MicroMill*, a computer steered drill mounted on a microscope. Samples were taken along the central growth axis, to avoid possible effects of evaporation during calcite deposition (Fairchild et al., 2006). For all samples, 300 µm tungsten carbide

dental drill bits from *Komet* were used. In function of the growth rate, 1000, 500 and 250 µm sampling resolutions were applied. For sample locations, see Fig. 2D. Samples were kept at 50°C prior to analysis. $\delta^{13}$C and $\delta^{18}$O isotope measurements were performed on a *Perspective* IRMS from *Nu Instruments*, coupled to a *Nucarb* automated carbonate preparation system and a minor amount (<100) on a *Kiel III* device coupled to a *Delta plus XL* from *Thermo-Scienitfic*. Two samples of the in-house standard MAR-2(2), which has been

calibrated against the international standard NBS-19 (Friedman et al., 1982), were measured every 10 samples to correct for instrumental drift. Reported values for the MAR-2(2) are 0.13 ‰ VPDB for $\delta^{18}$O and 3.41 ‰ VPDB for $\delta^{13}$C. Analytical uncertainties on standards from individual batches were ≤ 0.05 ‰ for $\delta^{13}$C and ≤ 0.08 ‰ for $\delta^{18}$O on the *Nu instruments* setup. Every eight samples a double was measured in a different batch to check for the reproducibility of the analytical method. Outliers were manually detected, removed and re-measured if

sufficient material was present. To check for isotopic equilibrium conditions during speleothem formation, nine Hendy tests (Hendy, 1971) consisting of 10 measurements, five at each lateral side, were carried out. In addition, six 30 µm thin-sections were taken along the growth transect (Fig. 2D).



## 4   Results

### 4.1   Speleothem morphology

Figure 2C -2E show the interpretation of the internal morphology of the Han-9. In the lower part of the speleothem, up to 365 mm dft, layering is extremely good expressed and consists of sub-millimeter sized alternations of whiter and slightly darker calcite. At the very base, some fine, brown detrital laminae can be seen, although they are confined to the lateral sides of the stalagmite. Layering is visible up to 365 mm dft. The calcite becomes progressively coarser and layering less expressed from around 430 mm dft onwards, where alternations
of thicker parts of denser and coarser calcite are present. Starting from 365 mm dft, the speleothem calcite has a very coarse appearance and layering is almost indistinguishable. This goes on until 304 mm dft, where a first discontinuity in growth, D1, appears. After D1, 100 mm of speleothem is characterized by alternating bands of dense and dark brown calcite with coarser, white calcite. Sometimes, subtle fine layering can be observed, mainly in the coarser parts or on the lateral sides. Between 200 and 176 mm dft, a band of dense, brown calcite is
present. Within this band, very fine laminae of red-brown material can be observed. This band ends with a second discontinuity, D2. Stalagmite growth of the upper part starts with a growth axis tilted towards the right, however after 20 mm the axis recovers to its original upright position. The entire upper section consists of dense and dark brown calcite, with little variation except for a coarser interval between 58 and 40 mm dft. Besides some subtle unconformities marked by the dashed lines, no internal layering is present. The location of the thin-
sections was chosen in function of the typical morphology displayed in the stratigraphic log in Fig. 2E. Fabrics are described according to Frisia (2015). In all thin sections, the dominant fabric of the calcite crystals was columnar (Fig. 3A). The layering, although often very well displayed macroscopically, was not distinguishable within the thin sections. Variations in fabric occur between macroscopically defined 'denser' and 'coarser' calcite (Fig. 2E) where the latter has smaller columnar calcite crystals with significantly more inter-crystalline
porosity often filled with fluid inclusions (Fig. 3B) and thereby described as columnar open. The coarser morphology has substantially larger crystals with less pore space and can be defined as columnar elongated. Another type of fabric occurs within the dark brown band of dense calcite between 200 and 176 mm dft. There, the columnar fabric is replaced with smaller more equant calcite crystals (Fig. 3C). This also covers D2 and shows that the nature of the discontinuity is actually a fine layer of brown detrital material.

### 4.2   U/Th dating


The results of the U/Th datings are shown in Table 1. Dating samples are labeled by "DAT-X", with X representing the sample number in Fig. 2D. All ages are displayed as "year BP", with 2015 CE as "Present". Errors are given as 2σ and range between 0.22 and 0.66 %, corresponding to 666 years and 212 years, respectively. The U and Th concentrations determined in the 2013 samples allowed reduction of the sample size,
resulting in smaller errors for the 2015 samples. From 672 to 176 mm dft, all ages are stratigraphically consistent; no age inversions occur when taking into account the 2σ error of the U/Th ages. Between 176 and 0 mm dft, the distribution of the ages is more chaotic, with the occurrence of several age inversions and outliers.

### 4.3   Stable isotopes: $\delta^{13}C$ and $\delta^{18}O$

Figure 2A and B show the results of the $\delta^{13}C$ and $\delta^{18}O$ analyses, plotted against sample depth in mm dft. All
values are expressed in ‰ VPDB. The $\delta^{13}C$ varies between -3.58 and -10.30 ‰, with an average of -7.53 ‰. The $\delta^{18}O$ values shows smaller variations, between -5.04 and -7.02 ‰, and averages at -5.91 ‰. Overall,



variations of both $\delta^{13}C$ and $\delta^{18}O$ seem to be smaller in the lower part of the stalagmite, and larger amplitude variations are present from around 400 mm dft upwards, where also distinctive changes in morphology occur.

To check whether the speleothem calcite was formed in isotopic equilibrium with the drip water, nine Hendy
tests were carried out over the entire stalagmite (Fig. 2D). The results are shown in Fig. 4. First of all, $\delta^{18}O$ tends to be rather variable along a single growth layer, with variations up to 0.3 ‰. Nevertheless, this is still well below a threshold value of 0.8 ‰ (Couchoud et al., 2009; Gascoyne, 1992). Further, no significant covariation between the $\delta^{13}C$ and $\delta^{18}O$ records along the layers can be observed, except maybe slightly for Hendy test 8 and 2. However, the most valuable indication that the speleothem calcite was formed under (at least) near-
equilibrium conditions is given by a very low R² of 0.0387 between the $\delta^{13}C$ and $\delta^{18}O$ variations along the central growth axis of the speleothem, pointing towards no significant covariance between the two stable isotope proxies (Fig. 2A and 2B).

### 5. Discussion

### 5.1 Age model

The StalAge algorithm (Scholz and Hofmann, 2011) was applied to the individual ages in order to construct an age-depth model, displayed in Fig. 5. For the U/Th dates obtained between 0 and 176 mm dft, errors had to be enlarged to 2000 years, to account for the occurrence of age inversions, resulting in increased errors calculated in the final age-depth model. This makes the model unreliable for this part, however, it is correct the assume a general time window for the growth of this upper part between ~104 and ~97 ka. Despite the problems with the
upper 176 mm of the stalagmite, valuable information could be retrieved from the age-depth model. It is clear that the stalagmite endured 3 separate growth phases, and that the discontinuities, expressed in the stalagmite morphology at 302 and 176 mm dft (Fig. 2D), correspond to two hiatuses separating these 3 growth phases. A second adjustment of the age-depth model had to be made at the end of growth phase 2. Morphologic evidence, being the dense, brown calcite with very fine laminae, points towards a decreased growth rate between 200 mm
and 176 mm dft, and thus there is no evidence that the age of DAT-19, taken at 179 mm dft, should be considered as an outlier. Even if a higher detrital Th content would disturb the outcome of DAT-19, this would result in an older age. Therefore, the result of the StalAge model was replaced with a linear interpolation between DAT-19 and DAT-16, taken at 201 mm dft (red line in Fig. 5).

The first and oldest growth phase starts at 125.34 +0.78/-0.66 ka with stable growth-rate of 0.02 mm yr$^{-1}$ up to
around 120.5 ka. After that, growth rate significantly increases, with values up to 0.15 mm yr$^{-1}$. At 117.27 +0.69/-1.02 ka, growth ceases and the first hiatus, H1, starts. The hiatus lasts 4.41 +1.10/-1.49 ka and at 112.86 +0.47/-0.41 ka growth phase 2 starts. DAT-4 and DAT-5 were taken 6 mm below and above the discontinuity and the age-depth model does not show any reason to question the timing of H1. As for growth phase 2, growth-rate remains at a constant pace of 0.04 mm yr$^{-1}$ until approximately 110.5 ka, where it decreases to 0.006 mm yr$^{-}$
$^{1}$. At 106.59 +0.21/-0.22 ka, the second growth phase ends. This hiatus (H2) lasts 3.0 +1.58/-1.28 ka, until 103.59 ka. Speleothem formation then completely comes to a hold at 97,22 +1.02/-2.61 ka. Given this age-depth model, stable isotopes were analyzed with an average temporal resolution of 16 years.

### 5.2 Interpretation of stable isotope proxies

Stable isotope time series are displayed in Fig. 6. The $\delta^{13}C$ values of modern $CaCO_3$ from the Han-sur-Lesse
cave average around 10‰ VPDB (Van Rampelbergh et al., 2014), 3‰ lower than the average for the Han-9.





Within the time series, changes up to almost 5‰ occur (Fig. 6). Such large changes in $\delta^{13}C$ are often attributed to shifts between C3 (arboreal) and C4 (grass) vegetation, since both types of vegetation leave a different $\delta^{13}C$ imprint in the soil $CO_2$. This leads to more negative values for $\delta^{13}C$ in cave carbonates of a C3 dominated vegetation (-14‰ to -6‰) compared to C4 vegetation (-6‰ to -2‰) (McDermott, 2004). Indeed, pollen records

for the 130-100 kyr period obtained from lake cores taken in a Maar lake in the Eifel, Germany (Fig. 1A and Fig. 7) indicate that increases of grass-like C4 pollen up to 40% of the total assembly occurred during that time period (Sirocko et al., 2005). Shifts in C3-C4 vegetation have even been observed the Vosges region, France, 300 km further south (Woillard, 1978; de Beaulieu and Reille, 1992; de Beaulieu, 2010). Based on these records and other pollen datasets for northern and central Europe (Helmens et al., 2014 and references therein) large $\delta^{13}C$

shifts in the Han-9 stalagmite could reflect changes in vegetation cover above the cave, controlled by climate changes. On the other hand, a study by Genty et al. (2003) has also shown that for temperate regions not only changes in C3-C4 vegetation determine the $\delta^{13}C$ signal, but also changes in vegetation activity. Less activity, or more specific less respiration by vegetation, results in smaller proportions of vegetation derived $CO_2$ relative to atmospheric $CO_2$ in the soil. This isotopically heavier atmospheric $CO_2$ would then cause an increase $\delta^{13}C$ of the

soil gas. This has also been attributed for $\delta^{13}C$ excursions on a centennial to annual scale in a Holocene speleothem from the Han-sur-Lesse cave (Van Rampelbergh et al., 2015), whereas on a seasonal scale $\delta^{13}C$ variations of speleothem calcite are attributed to Prior Calcite Precipitation (PCP) (Van Rampelbergh et al., 2014). Recent work from Schubert and Jahren (2012) has also indicated an additional control on $\delta^{13}C$ of C3 plants by photosynthetic discrimination. Recently, Wong and Breeker (2015) have shown that changes of up to -

2 ‰ over the LGM – Holocene period can be attributed to photosynthetic discrimination. This can also be taken into account, especially for time intervals were large changes in atmospheric $pCO_2$ are expected, such as the MIS 5e/5d transition, as evidenced by past atmospheric $pCO_2$ changes up to 100 ppm recorded in the Epica Dome cores in Antarctica (Petit et al., 1999).

In the mid-latitude setting of northwestern Europe, it is difficult to constrain the origin of speleothem $\delta^{18}O$

variations in terms of changes climate expression (i.e. temperature, precipitation), since it is known that several different processes, with variable influence, account for this variability (McDermott, 2004). A good overview of all processes possibly influencing the speleothem $\delta^{18}O$ is given by Lachniet (2009). One of the main processes acting on both precipitation $\delta^{18}O$ and calcite $\delta^{18}O$ is temperature. Temperature fractionation on vapor condensation was estimated to be around 0.6 ‰ °C$^{-1}$ (Rozanski et al., 1992) whereas the temperature dependent

fractionation between cave drip water and speleothem calcite for the Han-sur-Lesse cave was calculated to be - 0.2 ‰ °C$^{-1}$ (Van Rampelbergh et al., 2014). Combining these data gives a positive relation between temperature on speleothem $\delta^{18}O$. This has been attributed as one of the main drivers of $\delta^{18}O$ fluctuations in European speleothems (Boch et al., 2011, Wainer et al., 2013). It is very likely that in this record also the amount of precipitation partly influences the $\delta^{18}O$ signal of the speleothem. In the tropics, variations in monsoonal strength

are regarded as the main control on speleothem $\delta^{18}O$ via this amount effect (Wang et al., 2001) and changes in the amount of precipitation over time have also been considered as a driver for variations within European speleothem $\delta^{18}O$ records (Genty et al., 2003; Couchoud et al., 2009, McDermott et al., 2011). Temperature and precipitation controls are thus expected to contribute most to the speleothem $\delta^{18}O$ variability but as this record covers a interglacial-glacial transition, other processes acting on longer timescales should also be considered. A

significant contribution is to be expected from the variations in $\delta^{18}O$ of the source, i.e. the North-Atlantic ocean, because of fluctuations in global ice-volume. Waelbroeck et al. (2002) has estimated that during MIS 5d, average global $\delta^{18}O$ values were up to 0.5 ‰ higher compared to MIS 5e. Additionally, a study by Verheyden et al.





(2014) on a Holocene speleothem from the Han-sur-Lesse cave system has shown the important effect of kinetic processes overprinting the climate $\delta^{18}O$ signal, confirmed by a strong covariation of $\delta^{18}O$ with $\delta^{13}C$ and Mg/Ca and Sr/Ca ratio's as well. Similar covariance was observed for a 500 year record from the Common Era (Van Rampelbergh et al., 2015). In case of the Han-9, covariation between $\delta^{18}O$ and $\delta^{13}C$ is very low (Fig. 2A-B), so it is unlikely that kinetic processes acted on this speleothem during the LIG.

To conclude, in terms of interpreting the Han-9 $\delta^{13}C$ record, we can attribute increases up to several per mill on the glacial/interglacial and stadial/interstadial conditions to changes in vegetation type. Additional variability of the record on shorter time scales, during periods when no large changes in vegetation assembly are expected, is related to vegetation activity, in response to variating temperature and precipitation. Colder/dryer conditions would then lead to an increase in $\delta^{13}C$. This effect is can further be enhanced by PCP occurring during drier periods. The main driver of the $\delta^{18}O$ is temperature, although changes in the amount of precipitation will act on the $\delta^{18}O$ as well.

### 5.3 Climate in the Belgian area between 125.3 and ~97 ka

#### 5.3.1 125.3 ka: A late onset of the Eemian?

The modeled age for the start of the Han-9 growth is 125.3 ka (Fig. 5). Among the recent sampling missions for Belgian LIG speleothems, this is the oldest LIG sample found so far (S. Verheyden, pers. comm.). Although a flowstone from the same Han-sur-Lesse cave was reported to start growing at 130 +/- 10 ka (Quinif and Bastin, 1994), the accuracy of that one single alphaspectrometic dating result can be questioned. Cave systems in Belgium are known to be very sensitive recorders of glacial/interglacial changes, with speleothem deposition only during optimum interglacial conditions (Quinif, 2006). Strongest melting of the Greenland ice-sheet and reinforced AMOC conditions were present between 131.5 and 126.5 ka, as identified from the MD04-2845 core from the Bay of Biscay (Sánchez Goñi et al., 2012), and speleothem formation did occur in the Alpine region and in southern Europe before 125.3 ka (Mosely et al., 2015; Drysdale et al., 2009). This raises the question whether or not the start of Han-9 growth is just sample specific or if it represents a real, although maybe locally confined, climate event at 125.3 ka. The SCH-5 alpine speleothem was continuously deposited between 134.1 ± 0.7 and 115.3 ± 0.6 (Moseley et al., 2015). Within this record, an increase in the $\delta^{18}O$ proxy starting at 128.4 ka and lasting to 125.3 ka was identified as a warming phase. The warming phase recognized in this alpine speleothem occurs just prior the start of the Han-9 speleothem growth, emphasizing a possible link between the warming and the start of speleothem formation in Han-sur-Lesse. In the BDInf speleothem studied by Couchoud et al. (2009) from southern France (Fig. 1A), the interval between 125.3 and 123.8 was identified as a period of increased rainfall amount. A warmer/wetter period is potentially expressed in northwest European vegetation records as well such as the Eifel Maar record from Sirocko et al. (2005), located only 150 km from the cave site in this study. The 125 ka time is marked by the transition of a pollen assembly consisting mainly of pioneering *Betula* pollen with boreal *Pinus* towards an assembly significantly richer in thermophilous, broad-leaf tree pollen such as *Ulmus*, *Quercus*, *Corylus* and *Carpinus*. However, the chronology of this record was not constructed independently; the start of the Eemian s.s. was determined by cross-correlating with the U/Th dates of the SPA-50 Alpine speleothem record of Holzkämper et al. (2004) and set at 127 ka. The Han-9 $\delta^{13}C$ record at 125.3 has the most negative values for the entire 125.3-117.3 growth period, reaching almost -9 ‰. $\delta^{18}O$ on the other hand registers a decrease > 0.5 ‰ during the first 300 years of growth. This perhaps demonstrates that interglacial optimum conditions were already present before 125.3 ka, but that an increase in moisture availability caused by





enhanced precipitation above the cave, shown by the $\delta^{18}$O decrease, was the factor needed to trigger growth of the Han-9.

### 5.3.2 125-120 ka: Eemian optimum

The isotope records of the Han-9 are relatively stable between 125 and ~120 ka. The variation of $\delta^{18}$O seems to be sub-millennial and is largely confined between -5.7 and -6.3 ‰. The long-term trend, as displayed by a fitted 7-point running average (Fig. 7), shows lower variability especially when compared to younger growth periods of the Han-9 (i.e. between 120-117.3 and 112.9-106.6). Similar observations are made for the $\delta^{13}$C: sub-millennial variability restricted between -7 and -8 ‰, with the exception of a positive excursion towards -6 around 122 ka, and generally more stable than in other time intervals. During 125-120 ka, other paleoclimate records display stable interglacial conditions, such as speleothems from the Alpine region (Meyer et al., 2008; Moseley et al., 2015) and from Italy (Drysdale et al., 2009), and other archives including ice cores (NEEM community, 2013). In marine records off the Iberian Margin, the 125-119 period was identified as an interval of 'sustained European warmth' (Sánchez Goñi et al., 2012), following a time of enhanced Greenland melting between 131.5 and 126.5 ka. We therefore attribute the stability of our records to the Eemian climate optimum persisting in the Belgian area as well. This is also supported by the constant growth rate (Fig. 5) and the speleothem morphology, displaying a sequence of layered calcite, with no significant change over that time (Fig. 2C-E).

### 5.3.3 120-117.3 ka: Inception of glacial conditions

From 120 ka onwards, an increase in $\delta^{18}$O of 0.5 ‰ is observed. This change in $\delta^{18}$O of the speleothem corresponds with an elevated growth rate, supported by a decrease in stalagmite diameter. No major changes in the $\delta^{13}$C are observed. Although both the age-depth model and the speleothem morphology evidence a faster speleothem growth, likely related to an increase in moisture availability within the cave, there is no evidence in the Han-9 $\delta^{18}$O record for an increase in precipitation. Due to the amount effect, enhanced precipitation would cause the $\delta^{18}$O signal to shift towards more negative values, which is not observed in the record. The increase in growth rate and thereby accompanied faster $CO_2$ degassing during speleothem formation is known to act as a possible kinetic control on the speleothem $\delta^{18}$O (Hendy, 1971; Lachniet, 2009), yet kinetic control by fast degassing would also result in positive changes of $\delta^{13}$C (Baker et al., 1997), which is not the case here. Another possible explanation could be a temperature rise causing the elevated $\delta^{18}$O signal, although such locally confined temperature increase seems unlikely, because no other records (pollen, SST) supports this hypothesis. Most likely, this increase in $\delta^{18}$O is not related to any local climate effects but reflects a more regionally signal, which could be the increase of the source $\delta^{18}$O of the North Atlantic ocean, since a rise of 0.5 ‰ is in good agreement with estimations of the source $\delta^{18}$O variability over the MIS 5e/5d transition from Waelbroeck et al. (2002). As a matter of fact, a study by Hearty et al. (2007) combining Last Interglacial sea-level evolution at 15 sites around the world shows a rapid descent towards a MIS 5d low stand at 119 ± 2 ka. This also favors the hypothesis that the speleothem $\delta^{18}$O between 120 and 117.3 reflects changing ocean source due to ice build-up.

A severe change in the $\delta^{13}$C is not observed until 117.5 ka, where a sudden increase towards -4 ‰ occurs. This 5 ‰ change takes place within 200 years and happens just before the first hiatus in this speleothem, suggesting that the cessation in speleothem growth is indeed caused by a climate event. As the increase in $\delta^{13}$C likely reflects changes in vegetation, such as an increase of C4 like plants or a decrease in vegetation activity linked to a changing (drying and/or cooling) climate. The age of this event, 117.3 ka, stands out in other studies as well.





First of all, in the NGRIP $\delta^{18}$O record it falls within what is identified as Greenland Stadial 26 (NGRIP Members, 2004). Although the signature of this GS may not be as clear as the younger GS 25 or 24, it

corresponds with the overall decreasing trend observed in the ice $\delta^{18}$O, and also recognized in the more recent NEEM ice core (NEEM community, 2013). The global character of this climate event around 117.3 is evidenced by similar changes in the North Atlantic Ocean. A high-resolution study by Galaasen et al. (2014) has found perturbations of the $\delta^{13}$C of benthic foraminifera in marine sediment cores, interpreted as a sharp decrease in NADW formation at 116.8 kyr BP, in contrast with the high NADW formation observed during the MIS 5e.

Such reductions in NADW led to changes in AMOC, resulting in a reduced ocean heat transport and eventually cooling the climate. This hypothesis is further evidenced by lower sea surface temperatures and the presence of IRD at that time (Irvali et al., 2012). The lower resolution in the SST record of core MD04-2548 retrieved from the Bay of Biscay (Fig. 1)  could explain its absence in this archive (Fig. 7). Other accurately dated European speleothem records mark a similar climate deterioration around the same time. Perhaps the most obvious

example is the study from Meyer et al. (2008), where a 3 to 4 ‰ drop in $\delta^{18}$O of four different flowstones from the Entrische Kirche cave (Fig. 1A) is observed between 119 and 118 ka. This large drop in speleothem $\delta^{18}$O is believed to be caused by a severe cooling, and was defined as the glacial inception at the cave site. Furthermore, a subtle depletion occurs in the HÖL-10 stalagmite (Fig. 7) and was correlated to the $\delta^{18}$O drop from Entrische Kirche (Mosely et al., 2015).  Closer to the Han-sur-Lesse cave, a similar event was also observed in the Eifel

Maar record. There it was identified as the 'LEAP' or the Late Eemian Aridity Pulse and defined by an increase varve thickness, loess and charcoal content together with a higher abundance of grass pollen within the assembly (Sirocko et al., 2005), which would explain the increase in $\delta^{13}$C of the Han-9. Yet, the timing of the LEAP predates the Han-9 event by ~1ka. Nevertheless, we suggest that both records registered the same event and that the offset in chronology can be caused by the tuning of the Eifel record or the uncertainty of the Han-9 age-depth

model at 117.3 ka.

### 5.3.4 117.3-97 ka: Stadial/Interstadial changes in the Early-Weichselian

After the first hiatus, growth starts again at 112.9 ka. During the second growth phase of the Han-9 stalagmite (112.9-106.6 ka), interesting differences occur compared to the earlier formed part of the speleothem. First of all, the variation of both $\delta^{18}$O and $\delta^{13}$C is much larger (Fig. 6). Secondly, changes in stalagmite morphology appear,

with alternations between dense, darker calcite and more white, coarser calcite (Fig. 2C-E). The $\delta^{13}$C curve of the Han-9 shows a long term trend of increase until a maximum of -4 ‰ is reached at 110 ka. Superimposed on this trend, (Sub-) millennial variability ranging between 2 and 3 ‰ is present. In contrast, within the $\delta^{18}$O a long-term trend is not as obvious, although a minimum of -6.8 ‰ is reached between 111 and 110 ka. Both minima in stable isotope proxies correspond well with the timing of GS 25 (111-109 ka) observed in the NGRIP record

(plotted on the GICC05 timescale), implying that the stable isotopes of Han-9 reflect the temperature decrease of the stadial, which is likely since higher $\delta^{13}$C is linked to a less active vegetation cover during colder periods (or more C4 vegetation) and lower $\delta^{18}$O is caused by lower temperatures. The timing in the Han-9 record is also in agreement with the GS 25 registered in the NALPS record, which is believed to be dominantly temperature dependent (Boch et al., 2011). The summer Sea Surface Temperature reconstructions for marine core MD04-

2845 show a distinct decrease of ~10°C (Sánchez Goñi et al., 2012) at the same time. The variability of $\delta^{13}$C and $\delta^{18}$O in the Han-9, predating GS 25 (112.9-111 ka) has an inverse relationship suggesting that it is mainly temperature controlled. It thus appears that for a general cooler climate state, the amplitude of variability tends to increase as well, compared to a more stable Eemian optimum (125-120 ka). The timing of the second hiatus (106.6-103 ka) is similar to that of the occurrence of GS 24 in the NGRIP curve and is also registered in the

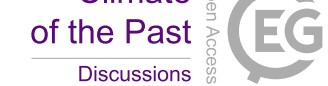

NALPS dataset from Boch et al. (2011). However, if the hiatus has any affinity with the GS 24, this raises the question why speleothem growth continued during the GS 25 period. A plausible explanation could be that growth never fully recovered from the GS 25, and that less favorable conditions (cooler/dryer) during the GS 24 interval were sufficient to cease growth. This assumption is grounded by decreased growth rate from 110 ka onwards (Fig. 5). In the Eifel Maar core, significant changes in vegetation occur from ~112 ka, with nearly all

pollen from broadleaf trees disappearing and the transition towards a pollen assembly dominated by coniferous trees and grasses (Fig. 7). Also, the time periods 110-108.5 ka and 106-104.5 ka are characterized by a significant increase in loess content and varve thickness, indicative for dryer conditions in that area and corresponding with the GS25 and GS24 intervals (Sirocko et al., 2005). In the third growth phase of the Han-9 (103.6 - ~97 ka), $\delta^{13}$C reaches its minimum, with a range between -8.5 and -10.5, which is far lower than the

average during the Eemian growth. This is remarkable, as pollen tend towards an assembly richer in grasses (Sirocko et al., 2005). However, it is difficult to make any further conclusions on this, as the age model is only poorly constrained below 103.6 ka (Fig. 5).

## 6. Conclusions

This study highlights the potential of Belgian speleothem proxies (i.e. growth, morphology and stable isotopes)

as recorders of regional and local climate change over the Eemian and Early-Weichselian in northwestern Europe. The start of speleothem growth occurs at 125.3 ka. At that time however, all of the European continental records are already within the Eemian climate optimum state. The $\delta^{18}$O record suggests that the eventual trigger starting speleothem growth was most likely moisture availability, linked to an increase in (local) precipitation at that time. Optimum Eemian climate conditions recorded in the Han-9 occurred between 125.3 and 120 ka, and

the stable isotopes and speleothem morphology indicate a relatively stable climate state. First signs of regional changing climate are observed in the $\delta^{18}$O proxy from 120 ka onwards, and are linked to a changing ocean source $\delta^{18}$O, caused by increasing ice volume. The end of the Eemian (and start of the Early-Weichselian) in the Han-9, at 117.3 ka, is preceded by a drastic change in vegetation activity and/or assembly that took place within 200 years, triggered by a decrease in moisture availability linked to a drying climate. This eventually led to

cessation of speleothem growth. This event appears to have a broad regional signature, as it is registered in other European records as well. However, pollen records imply that temperate vegetation seems to persist several millennia after 117.3 (de Beaulieu and Reille, 1992; Sirocko et al., 2005), giving to longer durations of the Eemian as defined in other records (Tzedakis et al., 2003). In addition, Han-9 also registered three Greenland Stadials occurring during the Early-Weichselian. The start of GS 26 is simultaneous with the end of the Eemian

at 117.3 ka, GS 25 occurs between 110.5 and 108.5 and GS 24 starts at 106.6. These chronologies are consistent with other European speleothem records. GS 25 is characterized by changes in vegetation (activity) caused by decrease in temperature and precipitation, whereas GS 24 is marked by a stop in speleothem growth. For the Early-Weichselian, local climate appears to be more sensitive during the early glacial conditions as the amplitude and frequency of variability tends to increase significant compared to the Eemian.

**Acknowledgements.** We thank the Domaine des Grottes de Han for allowing us to sample the stalagmite and to carry out other fieldwork. Ph. Claeys thanks the Hercules Foundation for upgrade of the Stable Isotope laboratory at VUB, and the VUB Strategic Research funding.




**Table 1**

| Sample Number | DFT (mm) | $^{238}$U (ppb) | $^{232}$Th (ppt) | $^{230}$Th / $^{232}$Th (atomic x10$^{-6}$) | $\delta^{234}$U* (measured) | $^{230}$Th / $^{238}$U (activity) | $^{230}$Th Age (yr) (uncorrected) | $^{230}$Th Age (yr) (corrected) | $\delta^{234}$U$_{Initial}$** (corrected) | $^{230}$Th Age (yr BP)*** (corrected) |
|---|---|---|---|---|---|---|---|---|---|---|
| DAT-10 | 668 | 531,5 ±0,6 | 1693 ±34 | 6077 ±122 | 622,5 ±1,5 | 1,1739 ±0,0016 | 124854 ±363 | 124805 ±364 | 885 ±2 | 124740 ±364 |
| **DAT-1** | **639** | **475,3 ±0,4** | **6419 ±128** | **1417 ±29** | **606 ±3** | **1,1608 ±0,0027** | **124781 ±653** | **124569 ±669** | **862 ±4** | **124506 ±669** |
| DAT-11 | 614,6 | 227,6 ±0,2 | 413 ±8 | 11096 ±223 | 704,0 ±1,6 | 1,2220 ±0,0013 | 122075 ±304 | 122049 ±305 | 993 ±2 | 121984 ±305 |
| DAT-12 | 581,5 | 289,7 ±0,2 | 1597 ±32 | 3563 ±71 | 675,2 ±1,5 | 1,1911 ±0,0014 | 120625 ±304 | 120542 ±310 | 949 ±2 | 120477 ±310 |
| **DAT-2** | **554** | **370,5 ±0,3** | **213 ±4** | **33652 ±692** | **661 ±3** | **1,1759 ±0,0029** | **119934 ±589** | **119925 ±589** | **927 ±4** | **119862 ±589** |
| DAT-13 | 484,2 | 390,1 ±0,4 | 1420 ±28 | 5246 ±105 | 637,7 ±1,6 | 1,1583 ±0,0015 | 119971 ±331 | 119915 ±333 | 895 ±2 | 119850 ±333 |
| DAT-14 | 426 | 337,7 ±0,3 | 243 ±5 | 26740 ±539 | 656,3 ±1,5 | 1,1679 ±0,0013 | 119179 ±289 | 119168 ±289 | 919 ±2 | 119103 ±289 |
| **DAT-3** | **397** | **373,4 ±0,3** | **681 ±14** | **10480 ±211** | **650 ±2** | **1,1596 ±0,0027** | **118564 ±539** | **118536 ±539** | **908 ±3** | **118473 ±539** |
| DAT-15 | 356 | 636,0 ±0,7 | 1406 ±28 | 8600 ±172 | 648,9 ±1,5 | 1,1533 ±0,0014 | 117661 ±306 | 117627 ±307 | 904 ±2 | 117562 ±307 |
| **DAT-4** | **311** | **226,6 ±0,3** | **1735 ±35** | **2526 ±51** | **667 ±3** | **1,1727 ±0,0036** | **118596 ±727** | **118481 ±731** | **932 ±5** | **118418 ±731** |
| **DAT-5** | **297** | **396,8 ±0,4** | **563 ±11** | **13324 ±269** | **677 ±2** | **1,1456 ±0,0027** | **112964 ±512** | **112942 ±512** | **931 ±4** | **112879 ±512** |
| DAT-17 | 271 | 213,5 ±0,1 | 1647 ±33 | 2460 ±49 | 692,3 ±1,4 | 1,1511 ±0,0011 | 112045 ±237 | 111930 ±251 | 949 ±2 | 111865 ±251 |
| DAT-18 | 238,5 | 239,0 ±0,2 | 208 ±4 | 21967 ±446 | 709,8 ±1,6 | 1,1594 ±0,0012 | 111337 ±254 | 111324 ±254 | 972 ±2 | 111259 ±254 |
| **DAT-6** | **214** | **216,2 ±0,3** | **939 ±19** | **4315 ±87** | **691 ±3** | **1,1365 ±0,0035** | **109887 ±637** | **109822 ±638** | **943 ±4** | **109759 ±638** |
| DAT-16 | 201 | 158,8 ±0,1 | 386 ±8 | 7695 ±155 | 680,1 ±1,5 | 1,1334 ±0,0012 | 110663 ±245 | 110627 ±246 | 929 ±2 | 110562 ±246 |
| DAT-19 | 179,5 | 179,8 ±0,1 | 3158 ±63 | 1064 ±21 | 711,7 ±1,5 | 1,1341 ±0,0011 | 107303 ±223 | 107042 ±289 | 963 ±2 | 106977 ±289 |
| **DAT-7** | **171** | **187,0 ±0,2** | **1051 ±21** | **3255 ±66** | **712 ±4** | **1,1089 ±0,0037** | **103615 ±671** | **103531 ±673** | **953 ±6** | **103468 ±673** |
| DAT-20 | 154 | 204,3 ±0,2 | 2749 ±55 | 1339 ±27 | 727,0 ±1,5 | 1,0929 ±0,0011 | 99868 ±215 | 99668 ±257 | 963 ±2 | 99603 ±257 |
| DAT-21 | 125,7 | 159,7 ±0,1 | 1709 ±34 | 1673 ±34 | 744,0 ±1,6 | 1,0861 ±0,0011 | 97373 ±206 | 97215 ±234 | 979 ±2 | 97150 ±234 |
| **DAT-8** | **98,7** | **149,3 ±0,2** | **3593 ±73** | **3593 ±73** | **737 ±4** | **1,1070 ±0,0039** | **100847 ±664** | **100772 ±666** | **980 ±6** | **100709 ±666** |
| DAT-22 | 56 | 196,9 ±0,1 | 978 ±20 | 3503 ±70 | 694,6 ±1,5 | 1,0554 ±0,0011 | 97680 ±206 | 97604 ±212 | 915 ±2 | 97539 ±212 |
| DAT-23 | 34 | 234,0 ±0,2 | 359 ±7 | 11228 ±226 | 668,5 ±1,6 | 1,0444 ±0,0013 | 98598 ±236 | 98574 ±237 | 883 ±2 | 98509 ±237 |
| **DAT-9** | **15,2** | **240,4 ±0,2** | **531 ±11** | **7898 ±161** | **677 ±3** | **1,0581 ±0,0039** | **99784 ±634** | **99750 ±634** | **897 ±4** | **99687 ±634** |

U decay constants: $\lambda_{238}$ = 1.55125x10$^{-10}$ (Jaffey et al., 1971) and $\lambda_{234}$ = 2.82206x10$^{-6}$ (Cheng et al., 2013). Th decay constant: $\lambda_{230}$ = 9.1705x10$^{-6}$ (Cheng et al., 2013).

*$\delta^{234}$U = ([$^{234}$U/$^{238}$U]$_{activity}$ − 1)x1000.  ** $\delta^{234}$U$_{initial}$ was calculated based on $^{230}$Th age (T), i.e., $\delta^{234}$U$_{initial}$ = $\delta^{234}$U$_{measured}$ x e$^{\lambda 234 \times T}$.

Corrected $^{230}$Th ages assume the initial $^{230}$Th/$^{232}$Th atomic ratio of 4.4 ±2.2 x10$^{-6}$.  Those are the values for a material at secular

equilibrium, with the bulk earth $^{232}$Th/$^{238}$U value of 3.8.  The errors are arbitrarily assumed to be 50%.

***B.P. stands for "Before Present" where the "Present" is defined as the year 1950 A.D.


**Table 1.** U/Th measurements of the Han-9 Stalagmite (University of Minnesota). Samples are arranged in stratigraphic order and those in bold indicate results acquired in 2013.



**Figure 1**

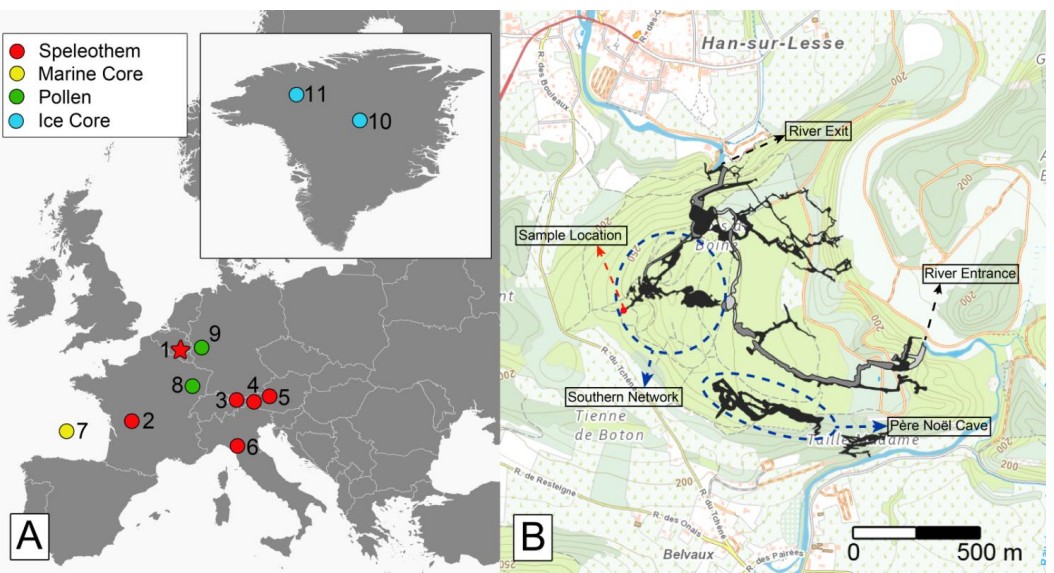

**Figure 1 A)** Location of the Han-sur-Lesse Cave site (Red Star) and other records mentioned in this study 1) Han-sur-Lesse; 2) Bourgeois-Delaunay Cave (Couchoud et al., 2009); 3) Hölloch Cave (Mosely et al., 2015); 4) Spannagel Cave (Hölzkamper et al., 2004); 5) Entrische Kirche Cave (Meyer et al., 2008); 6) Corchia Cave (Drysdale et al., 2009); 7) MD04-2548 (Sanchez-Goñi et al., 2012); 8) La Grande Pile (Woillard, 1978); 9) Eifel Maar (Sirocko et al., 2005); 10) NGRIP (NGRIP Members, 2004) and 11) NEEM (NEEM community, 2013). **(b)** Topographic map of the study area (source: NGI Belgium). The Han-sur-Lesse Cave system is plotted in black. The sampling site of the Han-9 is marked by the red dot. The Southern Network and the Père Noël cave are indicated with the dashed circles. Figure adapted from Quinif (2006).








**Figure 2**





**Figure 2**: Descriptive image of Han-9 **A)** $\delta^{13}C$ plotted against distance from top in mm; **B)** similar but now for
$\delta^{18}O$; **C)** High-resolution scan of the polished slabs; **D)** Interpretation of the internal structure of the speleothem.
Dashed lines = distinctive layers; Red lines = growth discontinuities; Grey areas = dating samples, numbers refer
to the samples in Table 1; Yellow line = central axis (sample axis); Blue boxes = thin section locations; Green
triangles = Hendy test locations; Brown lines = detrital material. **E)** Stratigraphic log: colors indicate the presence
of dense calcite (yellow) or coarse calcite (grey), more intense color = denser/coarser compared to lighter colored
parts. The visual expression of layering is indicated with the dashes in the log. Further description of the log can
be found in the figure and text

**Figure 3**

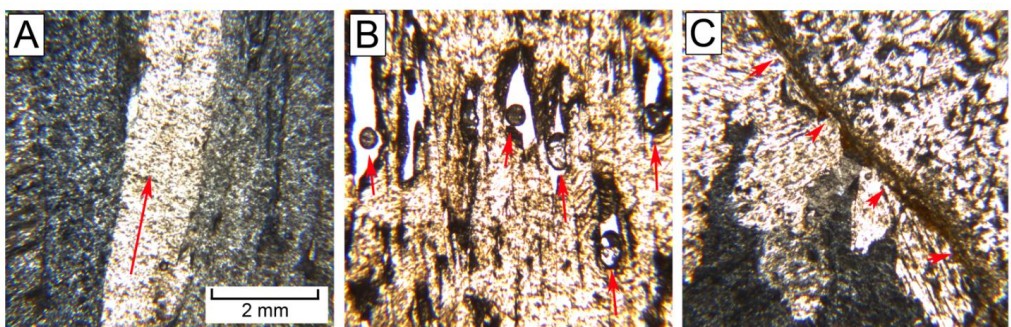

**Figure 3:** Thin section images of the Han-9. For locations, see Fig. 2D. All pictures are taken with crossed
polarized light, have the same scale and speleothem growth direction is upwards **A)** Thin section IIIB:
(Elongated) columnar calcite fabric. **B)** IIIA: Open columnar fabric, with fluid inclusions in the open voids **C)**
IIA: Discontinuity D2 is characterized by presence of brown detrital material. Also note the different fabric
underneath the hiatus.

     **Figure 4**

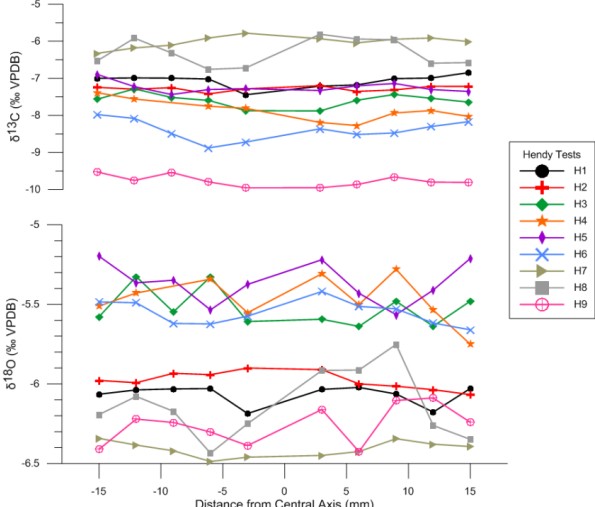

**Figure 4**: Results of the 9 Hendy tests carried out on the Han-9 stalagmite. Location of the tests can be found
in Fig. 2D.





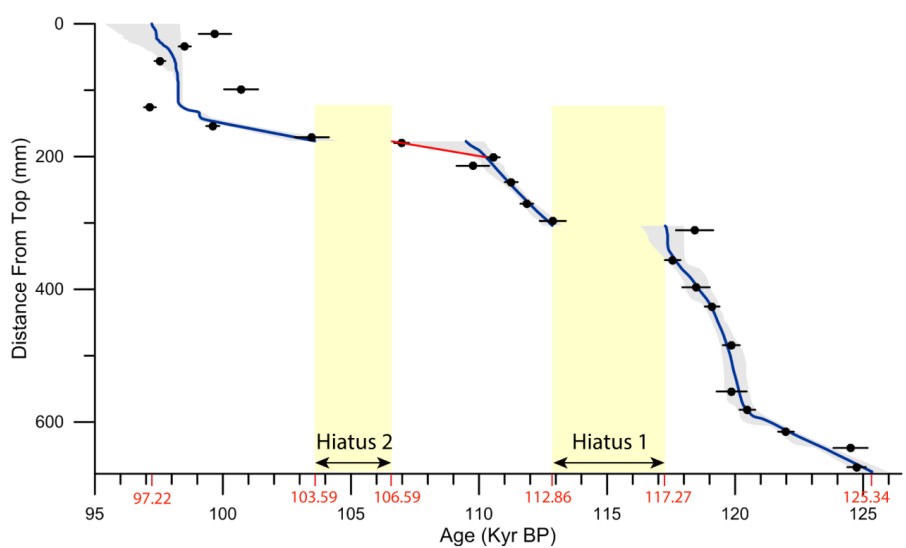

**Figure 5**: Han-9 age-depth model constructed with the StalAge algorithm (Scholz and Hoffmann, 2011). The actual age-depth model is represented by the blue line, the grey area marks the error. Ages in red represent important intervals and are discussed in the text.

**Figure 6**

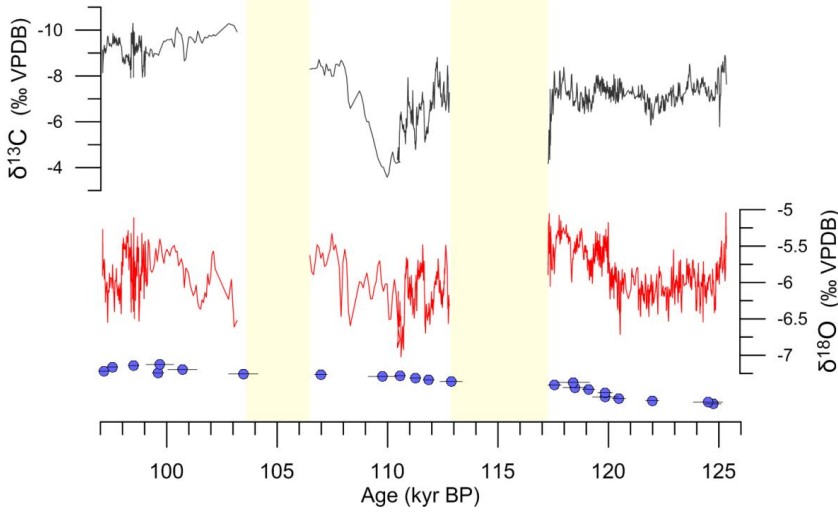

**Figure 6:** $\delta^{13}$C (black, inverted axis) and $\delta^{18}$O (red) time-series of the Han-9. In blue are the individual U/Th dates (see Table 1). Yellow bands mark the 2 hiatuses present in the speleothem.



**Figure 7**



**Figure 7:** Comparison of the Han-9 with other records in Europe, for locations, see Fig. 1A. **A)** June Insolation for 60°N (Berger and Loutre, 1991); **B)** Sea Surface Temperature reconstruction from MD04-2845 (Sanchez-Goñi et al., 2012); **C)** NALPS speleothem record (Boch et al., 2011); **D)** HOL-10 Austrian speleothem record (Moseley et al., 2015); **E)** NGRIP δ¹⁸O record with indication of Greenland Stadial intervals, plotted on GICC05 timescale (NGRIP members, 2004); **F)** & **G)** Han-9 stable isotope record with a 7-point moving average; **H)** U/Th dates of Han-9; **I)** Eifel Maar pollen assembly (Sirocko et al., 2005).



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
