# Peer review of "Paleoclimate in continental northwestern Europe during the Eemian and early Weichselian (125-97 ka): insights from a Belgian speleothem."

_Climate of the Past, 2016_

## Referee Comment (RC1) · Anonymous Referee #1 · 2 Mar 2016

**General Comments**

In this work, Vansteenberge and co-authors present growth, morphology, and carbon and oxygen records from a single Belgian speleothem. The authors created an age model using U-Th dates and then compared the data to Greenland ice-core records and European speleothem records. Speleothem studies from northern Europe are sparse; hence, the study has the potential to fill an important gap in the spatial coverage of speleothem records, and to make an important contribution to our understanding of northwest European paleoclimate during the last interglacial and glacial inception. The interpretation and reliability of the results is slightly limited by the analysis of only one speleothem, however, I appreciate that caves are not well decorated in Belgium and to harvest the few speleothems that remain in the hope of finding a replicate would be unethical. The manuscript fits within the scope of CP.

Overall, I support the publication of the work after some substantial revisions have been made. First of all, there are many technical issues associated with the use of language and in-house publication standards, but these can be easily rectified. More importantly, the approach taken towards some aspects of the age modelling needs further explanation, and the interpretation of the mechanisms controlling the d13C and d18O signals is unclear.

**Specific Comments**

Line 16 -  the larger European spatial coverage of last interglacial climate records does not need to come specifically from speleothems, consider revising.

Line 20 – Just because speleothems growing before 125.3 ka haven't been found yet, doesn't mean that they don't exist. Consider revising the bit about "speleothem formation starting relatively late in Belgium".

Line 30 –Greenland stadials are not recognized in the Belgian speleothem, rathermore, stadials occurred in Belgium that appear to be analogous to those in Greenland.

Line 31 – The last sentence needs revising. The second half related to Greenland Stadial 24 is fine, but the first half related to Greenland Stadial 25 is confusing. It is not clear that the Han-9 data is being referred to.

The first sentence of the introduction doesn't read well and needs revising. The higher temperature during the last interglacial also needs a reference.

Line 49 – the relationship between the two papers cited in this sentence is unclear.

Line 51 – I'm not sure that amelioration is the right word to use here

Line 59 – the orbital parameters were different to what?

Line 62 – D/O cycles don't control climate variability, they are a feature of it.

Lines 63-66 – Perhaps alternating would be a better word than succession? Nevertheless, the sentence needs revising. Atlantic Cold Events don't relate to the whole D/O cycle, plus the Atlantic Cold Events need a reference.

Line 70 – gradual cooling of what?

Line 95 – move the specifics about the length of the stalagmite to the cave setting. Then give more details about the sampling site. Average diameter? In situ or ex situ? Broken or complete? Refer to Fig 2C.

Line 108 – the rainfall is spread evenly throughout the year? Where does it come from? The same source all year? What is the modern d18O of the rainfall?

Line 114 – natural connection with which other parts of the cave? Perhaps indicate the underwater parts on the cave survey.

Line 113-114. Rephrase. The sample can't have been collected in the Réseau Sud OR the southern network of Han-sur-Lesse

Line 115 – mark where the tunnel is on Fig. 1B.

It would be helpful to mark the position of Réseau Rénversé, Réseau Sud, Gouffre de Belveaux, Trou de Han on Fig. 1B

The information about where the cave floods is confusing. Does flooding affect the sampling site?

Line 120 – when did the temperature logging take place? What were the sampling intervals?

Line 121 – why does the cave temperature reflect the average temperature for 2013, rather than the 1999-2013 average? Does this have implications for the interpretation later on?

The important point here is that the sample comes from the part of the cave with the stable temperature. Was that likely the same in the past? Or is there evidence for other entrances in the past that may have affected ventilation in this part of the cave?

Line 137 – normally the half-lives are given as part of the U-Th data table. If you wish to give them in the text, why only give $^{230}$Th and $^{234}$U, and not the half-lives for the other relevant isotopes?

Line 138 – specify atomic or activity ratio

Line 139 – And also Shen et al., Geochimica et Cosmochimica Acta 99 (2012) 71–86

What is the age datum?

Where were the stable isotopes measured?

Line 144 - 300µm diameter, radius, length?

Line 145 – Revise sentence beginning "In function of the growth rate…". It is unclear what is meant.

Line 146 -  why were samples kept at 50°C

Line 149 – what is the in-house standard made from?

Line 153  - do you mean replicate sample?

Line 154 – how are outliers defined?

Line 163 – it looks like the detrital material is on the base and not just the sides.

Line 163 onwards - it would help to be more specific about the sections of the speleothem being discussed. E.g. layering is visible between 365-700 mm dft.

How are the discontinuities identified? Macroscopically? By ages?

Line 174 - ...dashed lines in figure 2D....

Line 174 – do you mean no macroscopically visible internal layering is present?

Lines 178-181 – You say that the latter fabric (i.e. the coarser one) has smaller columnar calcite crystals, thus you describe it as "columnar open". You then contradict yourself and say that the "coarser morphology has substantially larger crystals.....defined as columnar elongated".

Line 183 – How can the smaller more equant calcite crystals also cover the fine layer of brown detrital material that is D2.

Later in section 5 there should be some discussion about why the morphology changes throughout the stalagmite. Why does the fabric change from dense to coarse? Why is there visible layering in some parts and not others. What might cause the growth axis to shift for 20mm?

Line 187 – The datum is given here as 2015 CE. Yet in the table it is 1950 AD.

Include some text in section 4.2 about the concentration and cleanliness of the samples.

Line 197 – be more specific about the section of stalagmite

Do you have any idea why there are age inversions?

Line 198 - What are the distinctive changes in morphology?

Line 199 – I think there should be some mention that the best test for isotopic equilibrium is to have a reproducible record (see Dorale and Liu, 2009). However, in the absence of a second speleothem, the Hendy test has been used instead. Do any of the modern monitoring studies record calcite deposition in isotopic equilibrium with the drip water?

Age model – why does the oldest part of the age model (figure 5) appear to miss out sample DAT-1? It is within stratigraphic order, yet the age model appears to completely miss it.

Lines 218-223 – Why does the age model need to be adjusted at the end of growth phase 2? Looking at the ages in table 1, DAT 6, 16 and 19 are all in stratigraphic order within the limits of dating uncertainty. Why then is there a discussion about whether DAT-19 is an outlier or not? Why has linear interpolation been applied at the end of growth phase 2? For this discussion, the names of the relevant samples should be added to figure 5.

Incidentally, it would be helpful to mark on one of the figures, either figure 2 or figure 5, what is meant by growth phases 1,2 and 3.

Lines 224-232 – Refer to figure 5 for the timing of growth phases. Where is the growth rate data plotted? What is the reason for the change in growth rate? What is the temporal range of the stable isotope sampling interval (i.e., not just the average).

Line 235 – what is the reason for difference in d13C between modern and Han-9?

Lines 236-258 – This section on d13C needs some revision. The shift of 5‰ **to enriched values** in Han-9 is centred on c. 110 ka. How long does this isotopic excursion last for? Be more specific about when the grass assemblage at Eifel was at 40%. How reliable is the chronology on the maar lake record? After much discussion about the different factors that control d13C in speleothems, there is no decision as to which mechanism(s) are controlling d13C in this particular record. The end of section 5.2 concludes by saying that shifts in d13C are caused by changes in vegetation type. After assessing all the different causes of changes in d13C, why has this conclusion been reached? i.e., Why is prior calcite precipitation discarded? Furthermore, why is there a mismatch between the pollen record from Eifel and the d13C of Han-9 if changing vegetation is the control on d13C? Between c.126-114 ka broad leaf trees dominate the assemblage, hence one would expect a speleothem d13C of -6 to -14 ‰, and this is the case at c.-7‰. The enrichment in speleothem d13C is then in agreement with the decline in broad leaf trees and increase in grass assemblages. This is fine too. But then the speleothem d13C shifts to c. -9 ‰ after 109 ka, i.e. it is even more depleted than it was between 126-114 ka, yet at this time grasses are most abundant (60%) and trees much less (40%). If the shifts in speleothem d13C are really reflecting changes in vegetation, then the absolute d13C values and pollen assemblages from the maar lake are not in agreement.

Line 260 – "precipitation" is a bit vague in terms of control on d18O. Perhaps use moisture source, amount effect.

Lines 272-273 – "precipitation controls" is too vague. Do you mean amount? Is the North Atlantic Ocean the source for the precipitation all year round? And would it have been the source in the past?

Line 276 – global d18O values of what?

Lines 283-288 – be more specific about which sections of the d13C signal are controlled by the different mechanisms.

Lines 296-297 – The results presented here show quite clearly that speleothem deposition is not restricted to optimum interglacial conditions.

Lines 316-317 – Be specific about why this demonstrates that interglacial conditions were already present at 125.3 ka.

Line 321 – refer to figure at end of first sentence

Line 324-326 – revise sentence so that the time period that is being referred to is clear

Line 329 – refer to figure 7 at end of list about speleothems and ice cores

Line 331 – reference for enhanced Greenland melting

Line 332 – you say there is a constant growth rate, but there are only three data points through which the age model passes over nearly a 9 cm interval. I come back to the earlier comment that the age for DAT-1 is completely ignored by the age model, despite being in stratigraphic order. If the age model took this age into account, then the growth rate would not be constant over the interval 120-125 ka.

Line 333 – no significant change in what over time? What is it about the layered calcite that indicates a stable climate?

Line 336 – refer to figure at end of first sentence.

Line 336 – don't use "onwards", be specific about the time interval being discussed. This is important for the sentence where it is stated that no major changes in d13C are observed. Major changes take place at 117.5 ka (as is discussed in the next paragraph), and this is "onwards" of 120 ka.

Line 336-337 – This needs rewording. At 120 ka, there is indeed an enrichment in d18O of 0.5‰, and the growth rate does change at this point, but the reduction in diameter doesn't appear to happen until later within this growth period. Also refer to figure 2 regarding growth diameter.

Line 336-352 – In the results section there was a lot of text about the d18O being controlled predominantly by temperature, but in this section it has now switched to both amount and source effect. Also, why does it switch from amount to source?

Line 355 – the sentence beginning "As the increase…." isn't complete. It is only the first part of a thought. What is happening as a result of the increase in d13C?

Line 358-361 – yes the beginning of the hiatus takes place within GS 26, but the age 117.3 ka doesn't stand out as a particular event within Greenland. GS-26 began at 119.1 ka (Rasmussen et al., 2014)

Line 370-371 – add the Meyer et al 2008 data to figure 7.

Hiatus 1 – boreal forests recover after the LEAP event, but the speleothem doesn't start growing until much later. Please comment.

Line 390 – use the GICC05modelext timescale rather than GICC05

Line389 – in Rasmussen et al 2014, GS 25 is 110.6 to 108.3 ka

Line 396 – there is a different timing here for GS25 compared to line 389

Line 396 – suggesting what is mainly temperature controlled?

Line 400-401- Rephrase the question. You ask why speleothem growth continued during GS 25, but then answer why it stopped during GS24.

Line 404 – Growth rate decreased at the same time d18O increased (amount effect=less rainfall), but d13C decreased

Line 411-412- weak ending. Yes the age model is poorly constrained, but the ages clearly cluster within the interval 97-103.6 ka, hence, the more depleted d13C appears to be in good agreement with the high abundance of grasses. Perhaps it is a problem with the Eifel Maar chronology, rather than the U-Th chronology? You should discuss whether PCP or fractionation effects might be responsible.

**Technical corrections**

**General comments – There are many issues related to grammar, use of articles, prepositions, language, and in-house standards. Please address them.**

Line 12, 13, 16, 20, keywords, 37, 42, 85, 350,   – "Last interglacial" is not a formal geological subdivision and hence should not be capitalized. See guidelines for authors. http://www.climate-of-the-past.net/for_authors/manuscript_preparation.html

Line 17, 91, 98, 99, 106, 113, 114, 125, 251, 265, 278294, 374 – Hans-sur Lesse Cave (author guidelines, Capitalize generic geographic terms, such as "river", when they are part of a place name,)

Line 18 - remove "The" before Han-9

Line 22 – …time; therefore, it … or …time. Therefore, it …

Line 24 - …ka, however, at

Line 26 -  define MIS

Line 28 - …that the climate …

Title, line 29, 46, 48, 79,82, 97, 381, 415, 422, 429, 433,  – again Early Weichselian is not a formal subdivision, it should be "early Weichselian"

Line 30 – "Greenland stadials" (lowercase)

Line 31 – … are recognized in Han-9 and the …

Line 35 – Keywords. millennial-scale variability, Greenland stadials

Line 37, 60, 62 – last glacial (lowercase)

Line 39 – provide abbreviation after Marine Isotope Stage

Line 42/43 - …about the LIG nomenclature…

Line 50 – Eem River valley

"Capitalize generic geographic terms, such as "river", when they are part of a place name, but do not capitalize the generic term when it appears on its own, when it follows a capitalized generic term, or when it is in the plural (e.g. Mississippi River, Mississippi River basin, Mississippi and Missouri rivers)." http://www.climate-of-the-past.net/for_authors/manuscript_preparation.html

Line 52 – ….vegetation to today.

Line 53 – ….of up to 20 ka…

Line 57 - …LIG has gained…

Line 60 - …Revise sentence beginning "Subsequent to the Eemian".

Line 60, 62 – lowercase "last glacial"

Line 62 – millennial-scale

Line 64 – …Greenland stadial (GS) and interstadial (GIS) phases….

Line 65 - ….Atlantic cold events…..

Line 66 – lowercase stadials

Line 66 – lowercase ice-rafted debris (IRD)

Line 67 – North Atlantic Ocean

Line 75 –lowercase interglacial

Line 76 – late Quaternary

Line 76 -78 – Revise grammar and prepositions

Line 92 – …and demonstrated that…

Line 95 – 70cm long (Under the author guidelines, there is no space between the number and the unit)

Line 100 - ~10km

Line 104 - …consisting of middle to late Givetian reefal….

Line 104, 111, 115, – Lesse River (author guidelines, capitalize river when part of a name)

Line 105 – (Fig. 1B)

Line 105 – 40m.

Line 106 – 200km and 200m

Line 106 – define a.s.l.

Line 106 – why swap from ~ to ca.

Line 108 – 10.2°C

Line 109 - …consists of C3…

Line 110 – Quercus trees, and

Line 112 – the sample has a different name

Line 120 – Short-term temperature…

Lines 121, 122, 124, 135 – remove space between number and unit

Line 136 – remove italics

Line 136 – it was already stated that the analyses were carried out at the University of Minnesota

Line 136 – define MC-ICP-MS

Line 142, 145, 147, 148, 149, 153 – remove italics

Line 143 – …growth axis to avoid…

Line 151 – define VPDB

Line 160 – Figures 2C-2E

Line 160 – of Han-9.

Line 161 – replace up to 365mm dft, with below 365mm dft

Line 161 – rephrase "extremely good expressed"

Line 161-162- …and consists of alternating sub-millimeter sized layers of white and slightly darker calcite.

Line 162 – Replace "At the very base " with something like "In the lower 10mm,…

Line 171 – Be more specific about the upper part. Perhaps, "Following D2, the axis of growth for the next 20mm is tilted to the right".

Line 176 – …..calcite (Fig 2E), where….    and    ….inclusions (Fig. 3B), and thereby described….

Line 189 – 666 and 212 years are the wrong way round

Line 194. Figures

Line 196 - … values show smaller….

Line 207 – Figs.

Line 213 - …it is correct to assume….

Line 216, 217 – spell three

Lines 224-232 –Remove spaces between numbers and units (see author guidelines). Revise text so that it is in the past tense, i.e., "The first and oldest growth phase started at 125.34 ….."

Using both first and oldest is unnecessary, one of them is redundant

Line 225 – After that, the growth….

Line 228 – As for growth phase 2, the growth….

Line 231 – comma instead of period in 97.22

Line 236 - …changes of up to….

Line 241 - …indicate that grass-like C4 pollen increased up to 40% of the total assembly between 106 and 98 ka.   (At the moment it is unclear which part of the 130-110 ka relates to the 40% grass assemblage)

Line 246 – Revise grammar in sentence about Genty, 2003.

Line 248 – … more specifically less …..

Line 249 – The isotopically heavier…

 Line 249 – …an increase in d13C of ….

Line 256 – change were to where

Line 257 – 100ppm

Line 260 - …of changing climate expression…

Paragraph beginning In the mid-latitude – check for spaces between numerals and units

Line 270 - …via the amount effect…

Line 274 - …covers an interglacial-glacial….

Line 275 – North Atlantic Ocean

Line 276 – global ice volume

Line 276 - Waelbroeck et al. (2002) estimated that…

Line 280 – plural ratios

Line 292- Revise first sentence.

Line 297 – Sentence beginning "Strongest melting….", restructure

Line 300 – spelling of Moseley

Line 300 – refer to Fig 1a for alpine and southern Europe speleothem studies

Line 303 – ka missing after 115.3 +- 0.6

Line 309 - …as well, such as

Line 310- Rephrase "The 125 ka time"

Line 313 – define s.s.

Line 321 - …records of Han-9 are…

Line 322 - …confined to between…

Line 324 – insert ka with dates

Line 324 - ….are made for d13C: sub-

Line 347 – ….more regional signal,….

Line 351 - …towards an MIS 5d….

Line 352 – insert ka after ages

Line 364 – define NADW

Line 364 - …observed during MIS 5e.

Line 366 - …cooling of the climate.

Line 374  - spelling Moseley

Line 375 - …an increase in varve thickness

Line 386 - …curve of Han-9 shows a long-term increasing trend to a maximum of -4‰ at 110ka.

Line 387 - ….within the d18O record a long-term…

Line 394 – sea surface temperature (lowercase)

Line 396 – in Han-9

Lines 410 – tends toward

Line 419 – recorded in Han-9

Line 420 – The first signs

Linr 430 – add ka to ages

Line 434 – amplitude and frequency of isotopic shifts

Figure 1 – spelling Moseley

Figure 1 – The sampling site of Han-9

Figure 2 – d18O plotted against distance from top in mm

Figure 5 – what is the error? The 95 5 confidence limit?

Figure 6 – spell two

Figure 7 – add Meyer et al., 2008. Use GICC05modelext timescale for Greenland. Separate F and G into d13C and d18O respectively. What are the horizontal lines on H?

---

## Referee Comment (RC2) · S. Vansteenberge et al. · 24 Mar 2016

Paleoclimate in continental northwestern Europe during the Eemian and Early-Weichselian (125-97 ka): insights from a Belgian speleothem By Stef Vansteenberge, Sophie Verheyden, Hai Cheng, Lawrence R. Edwards, Eddy Keppens, and Philippe Claeys

This manuscript provides a valuable contribution to our understanding of climate change in NW Europe during the Last Interglacial. The authors have done a thorough analysis of stalagmite Han-9 and worked to place this record in the context of other proxy records from Europe over this time interval. I feel that the manuscript should

be published following moderate revision. There are grammatical mistakes throughout that I have partially addressed in my enumerated comments. The authors should run a spelling and grammar check to identify additional errors that I did not address. Some of the key figures, particularly the age model and Fig. 7 require additional notations/interpretations to support the discussion and Fig. 6 is redundant (as the same data appears in Fig. 7) and should be deleted. I feel that the discussion regarding interpreting the stable isotope data could be more streamlined and the authors need to be consistent about how they interpret these isotopes. I have suggested adding interpretation notation onto Fig. 7 to help the reader (and the authors) understand the interpretations. Particularly the discussions of d18O are somewhat contradictory and inconsistent. The age model is also a source of concern, particularly after Growth Hiatus II. I suggest possibly eliminating (or substantially reducing) discussion of the isotope data after Hiatus II due to dating uncertainties. I suggest that the age model across Growth Phase II could also be improved. As for Fig. 7, the authors need to add a few additional records and reorganise to help with their interpretations and better support their discussion. I believe they have de-emphasised the Han-9 data in Fig. 7 and this should be remedied by reorganising the records and setting all Y-axes to the same length. With some additional work, this record could provide a valuable contribution to our understanding of European climate during the Last Interglacial.

General Comments

Section 4.1 Speleothem morphology: This entire section is rather unclear. The U/Th dates between 0 and 176 mm dft are unusable. I recommend not interpreting your record past Hiatus 2. I lack confidence in the Hendy Tests (see Dorale and Lu, 2009). The first potential problem is that the drill bit is 300 microns and the reported growth rates are 20, 40 and 150 microns. This suggests that at worst the authors are averaging 15 years and at best, 2 years. Another potential explanation for the very flat Hendy Test results for d13C is that these tests only extend to 15 mm from the central growth axis which is likely within the splash cup and not enough distance (even under conditions of kinetic fractionation) for degassing to occur as the drip progresses toward the flanks. Also, the variability in the d18O data along a (presumably) single growth axis is nearly 25 % of the total d18O variability through time. This is less than inspiring. I suggest that a better test for kinetic fractionation is to look at how d18O and d13C covary (or anticorrelate) over the length of the record. Superficially I did this and it appears that there are intervals of strong anticorrelation such as during the 'Eemian Optimum' and other periods of covariation such as after Hiatus II. Perhaps a running Pearson's could shed some light on when conditions favoured kinetic fractionation and this in itself provides valuable climate information. The presence of kinetic fractionation really only comes into play when aiming to apply the palaeotemperature equation. If this is not the goal (and I believe it is not here) then the presence or absence of kinetic fractionation does not rule out but can contribute information to the overall palaeoclimate interpretation. For the Age Model, the authors might want to try COPRA to see if it handles the hiatuses better. Did the authors run StalAge over the two growth intervals separately? I don't see why StalAge would interpolate incorrectly over the last 2 dates in Growth Phase II if it were run over the two growth intervals separately. I suggest that short of trying COPRA, the authors should rerun StalAge over Growth Interval II alone and see if the age model is more true to the U/Th dates and errors over that interval. I think that the current linear interpolation is unsatisfactory. Again, I would limit discussion of Growth Phase III due to dating uncertainties. Section 5.3 'A late onset of the Eemian' seems to be special pleading. Han-9 begins growing during the Eemian….in fact, well into it. I don't think there is enough evidence to support a late onset and would delete any of the discussion to this effect. Discussion of controls on Han-9 d18O: 1) Lower T will cause lower d18O due to rainout, distillation, etc. 2) ice build-up will sequester 16O, leaving behind a higher ocean d18O (source moisture) and driving rainfall d18O higher, 3) lower T would cause lower in-cave T and drive stalagmite d18O higher (Craig equation) but this effect is trivial. Effects 1 and 2 act against each other but one would win out (unless they cancel each other out entirely!). It is likely that lower T and increased rainout/fractionation will win, and rainfall (and thus Han-9 d18O) will be more

negative during colder conditions. It seems that the authors are arguing both sides in the manuscript but they need to pick one interpretation that applies to the whole record. It's interesting and somewhat puzzling that Han-9 growth seems to be restricted to intervals between Insolation max and mins but not during (i.e., no growth centred on 104, 115, and 127 kyr). It would be interesting to know if there is some explanation for this. The authors use 'ka' and 'kyr BP' in the manuscript. Select one and be consistent throughout. I find the match between Han-9 d13C and the pollen record particularly compelling at least until Growth Phase III when dating (and possibly the data) inaccuracies dominate. I disagree with stating that the start of Hiatus I marks the 'end of the Eemian', (such as the authors have done in the Conclusions and possibly elsewhere), rather, the authors could state that Han-9 ceases growth (Hiatus I) at 117.3 kyr, before the end of the Eemian as recorded by other proxies (references) suggesting that a critical threshold was reached in which conditions no longer favoured Han-9 deposition. This was possibly linked to a change in vegetation dynamics.

Enumerated Comments

Line 15: suggest changing to 'the Alps' rather than 'the Alpine region'. Change throughout. Line 20: change 'content' to 'composition' Line 22: delete 'the' before 'Han-9 growth' Line 27: change 'the speleothem d13C' to 'Han-9 d13C' and change 'a stop in' to 'cessation of' Line 28: change to 'suggesting a transition to significantly drier conditions'. Line 29: delete 'the' between 'both' and 'isotope' Line 30: reference to 'stalagmite morphology' in this sentence seems out of place. What about it? Line 38: change to 'were similar to or higher than those of the Holocene period and Present Day' Line 62: change 'scaled' to 'scale' Line 68-69: remove 'the' before AMOC and Atlantic Meridional Overturning Circulation. Line 76: change to 'Speleothems are ideal for' and 'the ability to construct accurate' Line 77: change to 'their potential to yield high resolution (up to seasonal scale) palaeoclimate records' Line 77: Change to 'Several stalagmite stable isotope proxies from Europe record optimum climatic conditions during the Eemian (Meyer et al, 2008; Couchoud et al, 2009) and D/O

climate events during the Early Weichselian (Bar-Matthews….'). Line 84: change to 'Belgium that expands the European' Line 109: change to 'consists of C3 type' Line 115: change to 'facilitating access' Line 129: change to 'provided a solid foundation for understanding' Line 134: change to 'age-depth model with additional sampling locations selected based on the preliminary age model and the…' Line 145: Change 'In function' to 'As a function' Line 153: Sentence beginning 'Every eight samples' is unclear. Rephrase this sentence. How was a double measured in a 'different batch' every 8 samples??? Line 161-2: Change to 'up to 365 mm dft, the calcite was well-laminated alternating between thick whiter and slightly darker layers. ' Line 163: change to 'stalagmite. From 365 mm dft, the calcite becomes progressively…' Line 174: Change to '..present. Thin-section locations were chosen as representative of the typical morphologies displayed in the …' See Comment for Fig. 3 Line 186: Change to 'The results of U/Th dating are shown' Line 196: Change to '….with an average of -5.91 ‰Line 197: Change to 'lower amplitude variability in both d13C and d18O occur in the lower part …' Line 198: Change to 'present from ∼400 mm dft upwards, corresponding to a distinct transition in morphology.' Lines 219-221: This discussion of why a linear interpolation is justified is unclear. Line 229: I suggest the authors revisit the age model over this interval as suggested in the general comments. This will likely change the growth rate reported here. Line 248: Change to 'more specifically, less respiration' Line 250: Change to 'This interpretation has also been attributed to d13C excursions…' Line 254: Change to '…discrimination, in which changes of up to ……discrimination (Wong and Breeker, 2015). Line 259: This is a very negative view of d18O in speleothems. I suggest rewording this sentence to something like "In mid-latitude Europe, several different processes (including temperature and precipitation) influence speleothem d18O variability (McDermott, 2004). Line 264: change 'whereas' to 'and' Line 266: Sentence starting with 'Combining these data…' does not make sense to me. Rephrase. Line 274: change to '…an interglacial-glacial transition, other processes acting on longer timescales (i.e., fluctuations in global ice-volume) should also be considered.' Line 283: Change to 'per mil' Line 284: Change to '…the

influence of glacial/interglacial and stadial/interstadial transitions on vegetation type.' Line 286: Change to '...is interpreted to reflect vegetation activity in response to changes in temperature and precipitation.' Line 305: Change to 'prior to the start' Line 306: Change to 'In the BDInf speleothem from southern France (Couchoud et al, 2009) (Fig. 1 A), the ...' Line 309: Change to 'Eifel Maar record (Sirocko et al., 2005), located only ...' Line 335: I would say 'gradual decrease in stalagmite diameter' because it is not particularly striking in Fig. 2. Line 338-9: Change to '...speleothem morphology support an increase in speleothem growth rate, possibly in response to an increase in ....' Line 355: Sentence fragment starting 'As the increase....' Rephrase. Line 367: I would add in here a mention of MD03-2664 once it is added to Fig. 7 Line 386: Change to 'long-term increasing trend until....' Line 387: Change to '..trend, sub-millennial scale variability ranging between....' Line 388: I would argue that the long-term trend is still visible in d18O over this interval and that d13C and d18O are strongly anticorrelated (something that your running Pearson's will reveal).The pattern of d18O and d13C variability will be more obvious when you expand the y-axes in Fig. 7. Again, I would highlight the time intervals you are discussing in the main text so that it is easier for the reader to cross reference. 'Both minima' –this is incorrect. D13C exhibits a maximum close to -4 per mil while d18O exhibits a minimum close to -7 per mil. Line 427: Change to '...resulting in a longer duration of the Eemian according to other records' Figures Figure 3: Figure 2d shows 6 thin sections. Which ones are shown here in A-C. Suggest labelling Fig. 2d more clearly so one can more easily cross reference Figs. 2d and 3 A-C. Figure 5: The section after Hiatus 2 is almost unusable. The error envelope shown (grey shading) does not cover the entire interval of age inversions and increased dating error. This needs to be expanded to cover the whole interval. I recommend excluding this growth interval from your discussions/interpretations. There is nothing obvious in Table 1 to explain why the dating is so imprecise over this interval. Also, in Figure 5, it would help if the authors would label the datapoints. The Age model discussion in section 5.1 refers to DAT-#'s quite a bit and it would help if the reader could make quick reference to a well-labelled

Fig. 5. Label Fig. 5 growth intervals and growth rates Figure 6: This figure is redundant considering that the same data are provided in Figure 7. I suggest deleting. In Figure 7 just darken slightly the non-averaged d18O and d13C curves…they are a bit too light to see right now. Figure 7: This figure is not very well laid out and could be vastly improved. The Han-9 data is minimised with compressed axes while the previously published datasets are at the forefront. Consider compressing the published data and expanding the axes on the Han-9 isotope data. I suggest making all of the axes the same length. It might be informative to move the Insolation record down over (or between) the Han-9 datasets. The U-Series datapoints should be plotted on 2 lines only (or one when they are not overlapping). The way they are now, suggests that there is some monotonic increase through time. Misleading. I suggest adding in the Yuan et al 2004 Dongge Cave data. I know this is a SE Asian Monsoon dataset but it tracks Greenland ice core data quite well and could help fill in some blanks. Add interpretations to figure. For instance, along axes like Han-9 d13C add down arrows (towards heavier d13C values) labelled with 'increasing C4 plants', 'Colder/drier', and 'increasing PCP', 'decreasing soil bioproductivity'. For Han-9 d18O add down arrows (towards lighter d18O values) decreasing ice volume Make certain that all axes are pointing in the correct direction regarding a uniform interpretation for the figure. The authors highlight the hiatuses but not the actual climate info. I suggest just labelling the data breaks as 'Hiatus 1' and 'Hiatus 2' (if the authors choose to keep growth phase III), but using shaded rectangles and more obvious labelling to highlight the climatic episodes discussed in the text (i.e., the Eemian climatic optimum from 120 to 125). Perhaps helpful to the reader would be to have the climatic periods identified (for instance in Paragraph 1 of the Introduction) in horizontal shaded rectangles at the bottom of the figure below the pollen data. In caption add d18O to 'NALPS speleothem record'. Add (SST) after 'Sea Surface Temperature' in B) if this is the first mention. Otherwise, just use SST. I suggest adding in the Irvali et al 2012 Fig. 3 b MD03-2665 Planktonic d18O record to Fig. 7, possibly overlapping with the Sanchez and Goni SST record. There is a lot more detail in the Irvali record and it is discussed in the text.

[Figure]

Please also note the supplement to this comment:
http://www.clim-past-discuss.net/cp-2016-15/cp-2016-15-RC2-supplement.pdf

―――――――――――――――――

---

## Author Comment (AC1) · 2 May 2016

The Authors wish to thank the two anonymous referees for the extended review of the discussion paper "Paleoclimate in continental northwestern Europe during the Eemian and Early-Weichselian (125-97 ka): insights from a Belgian speleothem" by Vansteenberge et al.

The general comments and technical corrections made by the reviewers will be taken into account when preparing the final version of the manuscript. In this reply, the authors would like to seize the opportunity to give additional information and corrections to certain issues suggested by the reviewers.

**1. Age Model**

To construct the age model, the StalAge algorithm of Scholz and Hoffmann (2011) was used. Although not mentioned in the original manuscript, the three growth phases were modeled separately, as suggested by reviewer #2. This appeared the best solution to handle the occurrence of the two hiatuses. Of course, we would like to stress that each model is an interpretation. Other algorithms such as COPRA are available and could give a solution to some of the issues encountered here, however, it could potentially create others. Originally, we experimented with MOD-AGE (Hercman and Pawlak, 2012) and COPRA (Breitenbach et al., 2012). With MOD-AGE, we encountered problems with the hiatuses. In the case of COPRA, the authors are not convinced of the idea of transferring age model uncertainties to uncertainties in the proxy values. Despite some caveats, the authors are still satisfied with the performance of StalAge in this study. Also, a comparison of different age models lies beyond the scope of this paper.

We improved figure 5 with the suggestions made by both reviewers, i.e. we added labels to the data points, included the growth rate and made a clear indication of the growth phases and hiatuses.

1.1 DAT-1 and the 125.3 – 117.3 ka interval

Reviewer #1 commented that DAT-1 is excluded from the age model, although it is in stratigraphic order. It is clear that DAT-1 has only limited weight in the actual model and an explanation for this can be found in the algorithm specifications (Scholz and Hoffmann, 2011). During the modeling process, the StalAge algorithm has a step where the data is screened for the occurrence of minor outliers and age inversions. This is done by fitting error weighted straight lines through subsets of three adjacent data points (Fig. 4 in Scholz and Hoffmann, 2011). However, DAT-1 is located in the basal part of the stalagmite, so less subsets of three data points can be used including DAT-1. If DAT-1 does not fit on the error weighted straight line created with the adjacent datapoints DAT-10 and DAT-11, which is the case here, the error of DAT-1 will be increased and the weight of DAT-1 in the Monte Carlo simulation for the age fitting will decrease. This results in less solutions where DAT-1 is included in the Monte Carlo simulated age models. The occurrence of substantial changes in growth rate in the boundary areas of a speleothem sample is recognized as a limitation of the StalAge algorithm (Scholz and Hoffmann, 2011).

1.2 DAT-19 and the start of the second hiatus (H2)

The StalAge model did not incorporate the DAT-19 date. This is again caused by the fact that DAT-19 is a date located in the boundary area of the stalagmite (it is the uppermost date of growth phase 2),

since the three growth phases were modeled separately. Here, we are convinced that DAT-19 should be included, as the stalagmite petrography shows clear evidence of a significantly decreased growth rate after DAT-16 (110.6 ka), i.e. very dense, brownish calcite with fine laminae. Also, a thin section showed smaller crystals. In complex cases, such as in this study where multiple hiatuses occur, the simplest model is still the best. Therefore, we believe that linear interpolation combined with good observations of changes in petrography, is a correct way to include DAT-19.

1.3  Growth phase 3

We agree with reviewer #2 that growth phase 3 has a bad dating potential. We already limited the interpretation of this growth phase in the original manuscript, but it is now clear to us that due to this bad chronology we should restrict the paleoclimate interpretation in this manuscript to the first two growth phases. That is why we also left out the stable isotope data of growth phase 3 in figure 7.

**2.  Correlation test of $\delta^{13}$C and $\delta^{18}$O**

As suggested by the second review, an additional test for correlation of $\delta^{13}$C and $\delta^{18}$O was done by calculating the Pearson's coefficient on the entire record and on the three growth phases separately (Table 1). However, we have to be careful because a correlation between $\delta^{13}$C and $\delta^{18}$O does not give conclusive evidence for the presence of kinetic fractionation, as both $\delta^{13}$C and $\delta^{18}$O are expected to be controlled by climate and could therefore show positive or negative covariation (Dorale and Lu, 2009). The Pearson's coefficients reveal that there is a clear difference between the separate growth phases. The first growth phase, with $\rho$ = 0.024, marks no covariation. The second growth phase, with $\rho$ = -0.467, has a substantial degree of negative correlation, whereas the third growth phase ($\rho$ = 0.461) has a positive correlation. We believe that the difference between the correlation coefficients of the separate growth phases indicate that different processes are controlling the stable isotope variability, and that the presence or absence of covariation reflects changes in climate conditions between the growth phases rather than the presence or absence of equilibrium.

**3.  Control on stable isotope variations in Han-9**

We agree with both reviewers that the discussion of the processes controlling the stable isotope variations in Han-9 is unclear and should be rewritten. Also, additional measurements of recent calcite were done to improve the understanding of the stable isotope functioning of the cave environment. Three samples of recent calcite were harvested by scraping off speleothem calcite from the tip of actively growing speleothems near the sample site of Han-9. The results are displayed in table 2.

3.1 $\delta^{13}$C

The $\delta^{13}$C in Han-9 is controlled by changes in vegetation assembly above the cave. This is deduced from the match between Han-9 $\delta^{13}$C and the abundance of grass pollen in the assembly of Sirocko et al. (2005) recovered from the Eifel maar. The agreement between Han-9 $\delta^{13}$C and the Eifel pollen assembly is remarkable; increases in $\delta^{13}$C occur when the percentage of grass pollen increases in the Eifel record. Also, the $\delta^{13}$C of recent calcite formed with a current forest-type vegetation cover above the cave is ~-8‰. These values are similar to those observed during the last interglacial in Han-9 (125.3-117.3 ka). In the original manuscript it is stated that changes in C3-C4 vegetation types control the $\delta^{13}$C variability in Han-9. We have to recall this statement because a higher abundance of grasses

does not necessarily result in an increased amount of C4 vegetation. First of all, C4 species make up only 1% of the total amount of vascular plant species in northwestern Europe today (Pyankov et al., 2010). Secondly, C4 species dominantly occur in a warmer, tropical climate (Ehleringer et al., 1997). Finally, within the subfamily of the Poideae, commonly referred to as the cool-season grasses and thriving in temperate European climate, all species use the C3 pathway (Soreng et al., 2015). The reason why speleothem calcite tends to be enriched in [13]C when vegetation is dominated by grasses is because grasses have a smaller biomass than trees and also the amount of soil respiration is lower, both leading to a smaller fraction of biogenic $CO_2$ compared to (heavier) atmospheric $CO_2$ within the soil (Genty et al., 2003). The similarity between the two records does not seem to hold up after 109 ka. From here on, $\delta^{13}$C becomes more depleted while Eifel record shifts towards an assembly dominated by grasses. This is especially the case after 106 ka, when grass pollen make up to 60% of the total assembly and where the most depleted values of $\delta^{13}$C, between -9 and -10‰, occur. These are even lower than the recent calcite $\delta^{13}$C values of ~-8‰. It is not clear to us what might have caused this depletion.

Prior calcite precipitation (PCP) is believed to control seasonal variations in the $\delta^{13}$C of Han-sur-Lesse speleothems, as concluded from an elaborate cave monitoring study by Van Rampelbergh et al. (2014). However, the study by Van Rampelbergh et al. (2014) was carried out on a large, tabular-shaped stalagmite with drip water discharge rates of 300mL min$^{-1}$ and growth rates of ~1mm yr$^{-1}$, so caution is required when extrapolating these cave monitoring conclusions to smaller, slower growing stalagmites in a different part of the cave system. If PCP occurs at one site in the cave, it does not mean that it occurs over the entire cave (Riechelmann et al., 2011). Since no additional data on Sr and Mg is currently available for Han-9, the presence of PCP cannot be confirmed, neither rejected.

To conclude, the control on Han-9 $\delta^{13}$C is the amount of biogenic $CO_2$ in the soil, caused by changes in the vegetation type above the cave (forest/grasses) and thus directly linked to climate. Lower, depleted $\delta^{13}$C values occur during warmer and wetter periods, when vegetation is dominated by temperate trees. Higher, enriched $\delta^{13}$C values of speleothem calcite correspond with a higher abundance of grasses above the cave during colder/dryer climate intervals.

**3.2  $\delta^{18}$O**

The $\delta^{18}$O variations in Han-9 are believed to reflect the temperature change, which controls the rainwater $\delta^{18}$O through temperature dependent fractionation on the vapor condensation in the atmosphere. Lower temperatures leads to more negative rainwater $\delta^{18}$O which is then reflected in the speleothem $\delta^{18}$O. This relation is clear in growth phase 2, where more positive $\delta^{13}$C values, which are known to reflect lower temperatures through vegetation changes, correspond with more negative $\delta^{18}$O. This is also reflected in the strong negative correlation (Table 1). However, this does not explain the increase in $\delta^{18}$O starting at 120 ka, as we would expect a decrease of temperature here. An increase in $\delta^{18}$O caused by a growing ice-sheet is a plausible explanation, as sea level reconstructions also indicated that regression, caused by ice build-up, started at that time (Hearty et al., 2007). During growth phase 2, temperature drops significantly, leading to a lower $\delta^{18}$O which cancels out the ice build-up effect.

**Response to minor comments**

Line 70: gradual cooling of the general climate state.

Line 95: The Han-9 was deliberately sampled because it had fallen over and was already broken into three parts. In this way, no other speleothems had to be destroyed. The speleothem was in-situ, no signs of transport near the sample site were observed.

Line 108: There is no distinct seasonal trend in the amount of rainfall over a year. The dominant moisture source in northwestern Europe is the subtropical North-Atlantic Ocean, and this remains constant throughout the year (Gimeno, 2010). The modern $\delta^{18}O$ of rainfall above the cave was monitored by Van Rampelbergh et al. (2014) over the entire year 2013 and seasonally varied between -17‰ in winter and -4‰ in summer.

Line 115: Flooding does not affect the sampling site.

Line 120: Temperature logging took place in the Réseau Renversé and sampling interval was 2 hours.

Line 121: The average temperature for the 1999-2013 period is 10.2°C, which is 1°C higher than the average temperature for 2013. The logging shows that the cave temperature is constant through the year and that the cave environment is sensitive to changes in the average temperature above the cave.

There is no evidence for any changes in the morphology of the Réseau Renversé since the last interglacial.

Line 138: The $^{230}Th/^{232}Th$ atomic ratio.

Line 139: The age datum is 1950 CE.

Line 142: Stable isotopes were measured in the Stable Isotope Laboratory at the Vrije Universiteit Brussel.

Line 144: Tungsten carbide drill bits with a diameter of 300μm were used.

Line 146: Samples were kept at 50°C prior to analysis to avoid contamination with atmospheric water vapor.

Line 149: The MAR-2 standard consists of Marbella Limestone.

Line 153: Every eight samples a replicate was measured in a different batch to check the reproducibility of the analytical method.

Line 154: A point was identified as an outlier if the difference of the value and the average of 10 previous and 10 following points was higher than two standard deviations of the 20 adjacent points.

Line 167: The discontinuities were identified macroscopically. Later on, these two discontinuities were attributed to hiatuses based on the ages.

Line 174: No internal layering is visible macroscopically.

Line 183: The smaller, equant calcite crystals occur between 200 and 176mm dft and are then followed by a fine layer of brown detrital material representing D2.

Line 185: Samples were pretty clean, as the detrital Th content, estimated by $^{232}$Th concentration and an initial $^{230}$Th/$^{232}$Th ratio of 4.4 ±2.2 x10$^{-6}$, is relatively low in all samples (range 6419 – 208 ppt). This leads to only minor corrections for the $^{230}$Th age (table 1 in original manuscript).

Line 198: Larger amplitude variations in $\delta^{13}$C and $\delta^{18}$O are present between 300 and 170mm dft. This is the speleothem section in between the two discontinuities and it is characterized by abrupt changes between dense, darker calcite and coarse, whiter calcite.

Line 199: Equilibrium deposition in Han-sur-Lesse Cave was observed by Van Rampelbergh et al. (2014).

Line 224-232: The temporal sampling interval was calculated based on the age model. The largest sampling interval is ~100 years, occurring just before Hiatus 2. The smallest interval is 0.3 years, which is located before Hiatus 1, between 330 and 320mm dft (growth rate curve in Fig. 5).

Line 291: The title "A late onset of the Eemian" is indeed misleading, as it is clear from other records and Han-9 as well that Eemian interglacial conditions were already present when the speleothem started growing.

Line 316-317: At the start of Han-9 growth, $\delta^{13}$C is depleted with values up to -9‰. Such depleted values are indicative for vegetation dominated by trees above the cave. This is also evidenced by the abundance of tree pollen in the Eifel record. Because of this forest type vegetation it is assumed that, in terms of temperature, interglacial conditions were already present before 125 ka.

Line 333: There are no changes in the morphology of the speleothem in between 125 and 120 ka, i.e. there is well-expressed layered calcite with constant layer thickness. It is believed that this is caused by absence of distinctive changes in temperature/water availability, as it corresponds to the period where the stable isotopes also show the smallest variations.

Line 355: Nothing happens as a result of the $\delta^{13}$C increase, it is the enrichment in $^{13}$C itself that is the result of an increase in grass-type vegetation caused by a drying/cooling event between 117.5 and 117.3 ka. Boreal forests do recover after the LEAP event, but the speleothem doesn't start growing until much later. A similar observation is made for the start of the speleothem growth at 125.3 ka: interglacial conditions were already present, yet the speleothem growth did not start immediately. It is not clear whether this is a site-specific effect or controlled by climate, but in the latter we could hypothesize that climate conditions need to be more favorable to initiate speleothem growth than to sustain it.

Line 396: The variability in the $\delta^{13}$C and $\delta^{18}$O between 112.9 and 111 ka is mainly temperature controlled: 1) lower temperatures lead to more negative $\delta^{18}$O values and 2) during colder periods grasses are more abundant causing an increase in $\delta^{13}$C. This negative correlation of $\delta^{13}$C and $\delta^{18}$O is reflected in the Pearson's correlation coefficient of -0.467.

Figure 5 is updated with the suggestions made by both reviewers.

Figure 7: As suggested by reviewer #2, we added three additional datasets to figure 7:

- MD03-2664 planktonic $\delta^{18}$O record (Irvali et al., 2012)
- Asian monsoon speleothem $\delta^{18}$O record (Yuan et al., 2004)

- TKS flowstone $\delta^{18}O$ (Meyer et al., 2008).

Axis of Han-9 $\delta^{13}C$ and $\delta^{18}O$ were enlarged, insolation curve moved closer to Han-9 stable isotope data, U/Th data axis adjusted. Speleothem records from the Alps were plot on a single axis. MIS5e and 5d subdivision was added. Also, the climatic intervals identified in Han-9 are now displayed in the figure. Arrows indicating control on stable isotope changes were added.

**Additional references**

Breitenbach, S. F. M., Rehfeld, K., Goswami, B., Baldini, J. U. L., Ridley, H. E., Kennett, D. J., Prufer, K. M., Aquino, V. V., Asmerom, Y., Polyak, V. J., Cheng, H., Kurths, J., Marwan, N.: Constructing Proxy Records from Age models (COPRA), Climate of the Past, 8, 1765-1779, 2012.

Dorale, J. A., Liu, Z.: Limitations of Hendy test criteria in judging the paleoclimatic suitability of speleothems and the need for replication, Journal of Cave and Karst Studies, 71, 73-80, 2009.

Ehleringer, J. E., Cerling, T. E., Helliker, B. R., C4 photoynthesis, atmospheric CO2, and climate, Oecologia, 112, 285-299, 1997.

Gimeno, L., Drumond, A., Nieto, R., Trigo, R. M., Stohl, A.: On the origin of continental precipitation, Geophysical Research Letters, 37, L13804, doi:10.1029/2010GL043712, 2010.

Hercman, H., Pawlak, J.: MOD-AGE: an age-depth model construction algorithm, Quaternary Geochronology, 12, 1-10, 2012.

Pyankov, V. I., Ziegler, H., Akhani, H., Deigele, C., Lüttge, U.: European plants with C4 photosynthesis: geographical and taxonomic distribution and relations to climate parameters, Botanical Journal of the Linnean Society, 163, 283-304, 2010.

Riechelmann, D. F. C., Schroeder-Ritzrau, A., Scholz, D., Fohlmeister, J., Spoetl, C., Richter, D. K., Mangini, A.: Monitoring Bunker Cave (NW Germany): A prerequisite to interpret geochemical proxy data of speleothems from this site, Journal of Hydrology, 409, 682-695, 2011.

Soreng, R. J., Peterson, P. M., Romaschenko, K., Davidse, G., Zuloaga, F. O., Judziewicz, E. J., Filgueiras, T. S., Davis, J. I., Morrone, O.: A worldwide phylogenetic classification of the Poaceae (Gramineae), Journal of Systematics and Evolution, 53(2), 117-137, 2015.

Yuan, D., Cheng, H., Edwards, R. L., Dykoski, C. A., Kelly, M. J., Zhang, M., Qing, J., Lin, Y., Wang, Y., Wu, J., Dorale, J., An, Z., Cai, Y.: Timing, Durqtion and Transition of the Last Interglacial Asian Monsoon, Science, 304, 575-578, 2004.

|  | ρ | # measurements |
|---|---|---|
| **GP1** | 0,024 | 599 |
| **GP2** | -0,467 | 235 |
| **GP3** | 0,461 | 284 |
| **Total** | -0,197 | 1118 |

**Table 1:** Pearson's coefficient of correlation (ρ) calculated for the three growth phases and for the total record.

|  | $\delta^{13}C$ | $\delta^{18}O$ |
|---|---|---|
| **RC1** | -8,19 | -5,73 |
| **RC2** | -8,20 | -6,25 |
| **RC3** | -7,80 | -6,48 |

**Table 2:** $\delta^{13}C$ and $\delta^{18}O$ analysis of three recent calcite samples from the Réseau Renversé.

[Figure]

**Figure 5 revised:** Han-9 age-depth model constructed with the StalAge algorithm (Scholz and Hoffmann, 2011). The actual age-depth model is represented by the yellow line, the grey area marks the error. Numbers represent the sample labels (Table 1 in original manuscript). The brown curve displays the growth rate. Numbers in red indicate important dates and are discussed in the text.

[Figure]

A  MD95-2042, Summer SST

B  MD03-2664, Planktonic δ18O

C  NGRIP δ18O
GS24   GS25   GS26

D  Dongge Cave δ18O

E  Alpine Speleothem δ18O
NALPS
TKS
Höl-10

F  Grasses
Conifers
Broad-leaf trees
LEAP
Eifel Maar % Pollen

G  June Ins. 60°N (W/m²)

H  Ice volume increase
Han-9 δ18O (‰ VPDB)
T° Decrease

I  Han-9 δ13C (‰ VPDB)
Increase in Grasses, Decreased Bioproductivity, Colder/dryer climate

LEAP

J  Hiatus 2 Stadial   Stadial   Hiatus 1   ice build-up   Eemian Optimum

MIS5d   MIS5e

Age (ka)
105   110   115   120   125   130

**Figure 7 revised:** Comparison of Han-9 stalagmite with other records. The shaded blue area marks the occurrence of the Late Eemian Aridity Pulse (LEAP) in the Eifel record and its equivalent in other records. **A)** Sea Surface Temperature (SST) reconstruction from marine core MD04-2845 (Sanchez-Goñi et al., 2012); **B)** Planktonic $\delta^{18}$O from marine core MD03-2664 (Irvali et al., 2012); **C)** NGRIP $\delta^{18}$O record with indication of Greenland Stadial intervals (NGRIP members, 2004); **D)** Asian monsoon reconstructions from Dongge Cave (D3 & D4 stalagmites; Yuan et al., 2004); **E)** Alpine speleothem $\delta^{18}$O (TKS: Meyer et al., 2008; NALPS: Boch et al., 2011; HöL-10: Moseley et al., 2015)**; F)** Eifel Maar pollen assembly (Sirocko et al., 2005); **G)** June insolation for 60°N (Berger and Loutre, 1991); **H)** & **I)** Han-9 stable isotope record with a 7-point moving average and U/Th dates; **J)** Paleoclimate interpretation of Han-9.

---

## Author Response (AR2)

[revised manuscript text omitted]

Comment [S1]: Structure of the stable isotope interpretation changed: subdivision between isotopic equilibrium, d13C and d18O. Isotopic equilibrium discussion was added since the Hendy tests are left out in the revised version. Here, the Pearson's correlation tests are discussed.

Comment [S2]: The entire section of δ13C has been modified according to the reviewers comments and the authors response.

[revised manuscript text omitted]

**Point-by-Point reply to the comments**

875    Both reviews treated separately. In black: suggestions made by the reviewers, in red: revised text, added text or comment made by the authors.

**REVIEW #1**
**Specific Comments**

880    Line 16 - the larger European spatial coverage of last interglacial climate records does not need to come specifically from speleothems, consider revising.
Changed to: To better understand regional climate changes over the past, a larger spatial coverage of European last interglacial continental records is essential and speleothems, because of their ability to obtain excellent chronologies, can provide a major contribution

885

Line 20 – Just because speleothems growing before 125.3 ka haven't been found yet, doesn't mean that they don't exist. Consider revising the bit about "speleothem formation starting relatively late in Belgium".
Changed to: The speleothem started growing relatively late within the last interglacial, at 125.3 ka, as
890    other European continental archives suggest that Eemian optimum conditions were already present during that time.

Line 30 –Greenland stadials are not recognized in the Belgian speleothem, rathermore, stadials occurred in Belgium that appear to be analogous to those in Greenland.
895    Changed to: Stadials that appear to be analogous to those in Greenland are recognized in Han-9 and the chronology is consistent with other European (speleothem) records.

Line 31 – The last sentence needs revising. The second half related to Greenland Stadial 24 is fine, but the first half related to Greenland Stadial 25 is confusing. It is not clear that the Han-9 data is
900    being referred to.
Changed to: Greenland Stadial 25 is reflected as a cold/dry period within Han-9 stable isotope proxies and the second interruption in speleothem growth occurs simultaneously with Greenland Stadial 24.

905    The first sentence of the introduction doesn't read well and needs revising. The higher temperature during the last interglacial also needs a reference.
Changed to: The last interglacial (LIG) period is known as the time interval before the last glacial period during which temperatures were similar to or higher than those of the Holocene period and Present Day (Otto-Bliesner et al., 2013).

Line 49 – the relationship between the two papers cited in this sentence is unclear.

Changed to: The term "Eemian" was originally introduced by Harting (1875) and was characterized by the occurrence of warm water mollusks in marine sediments of the Eem River valley, near Amsterdam, the Netherlands.

Line 51 – I'm not sure that amelioration is the right word to use here

Changed to: Nowadays, the Eemian is mostly interpreted as an interval of warmer climate associated with the spread of temperate mixed forests in areas with similar vegetation to today (Kukla et al., 2002).

Line 59 – the orbital parameters were different to what?

Changed to: Therefore, the LIG gained a lot of attention from both paleoclimate and climate modelling communities for studying a warmer climate state and potential future sea-level rise (Loutre et al., 2014; Goelzer et al., 2015), even though the present-day configuration of Earth's orbital forcing parameters is different (Berger and Loutre, 2002).

Line 62 – D/O cycles don't control climate variability, they are a feature of it.

Changed to: A major feature of climate variability during the last glacial is the occurrence of millennial-scale, rapid cold-warm-cold cycles, known as Dansgaard-Oeschger (D/O) events (Bond et al., 1993).

Lines 63-66 – Perhaps alternating would be a better word than succession? Nevertheless, the sentence needs revising. Atlantic Cold Events don't relate to the whole D/O cycle, plus the Atlantic Cold Events need a reference.

Changed to: These D/O cycles are expressed as alternating Greenland stadial (GS) and interstadial (GIS) phases in Greenland ice cores (Dansgaard et al., 1993; NGRIP members, 2004) and they also have affinity with Atlantic cold events registered in sea-surface temperature proxies (McManus et al., 1994).

Line 70 – gradual cooling of what?

Changed to: Nevertheless, according to Barker et al. (2015), it is more likely a non-linear response of a gradual cooling of the climate than a result of enhanced fresh-water input by iceberg calving, as previously proposed by Bond et al. (1995) and van Kreveld et al. (2000).

Line 95 – move the specifics about the length of the stalagmite to the cave setting. Then give more details about the sampling site. Average diameter? In situ or ex situ? Broken or complete? Refer to Fig 2C.

Added to section 2: Han-9, the stalagmite presented in this study, was deliberately sampled because it was already broken into three parts, so no other speleothems had to be destroyed. Although the sample was broken, it was still in situ. The candle-shaped stalagmite has a length of 70cm (Fig. 2C-E).

Line 108 – the rainfall is spread evenly throughout the year? Where does it come from? The same source all year? What is the modern d18O of the rainfall?

Changed to and addition of: . The amount of precipitation does not follow a seasonal distribution (Royal Meteorological Institute, RMI). The dominant moisture source in northwestern Europe is the North Atlantic Ocean, and this remains constant throughout the year (Gimeno, 2010). Modern δ18O of rainfall seasonally varies between -17‰ in winter and -4‰ in summer (Van Rampelbergh et al., 2014).

Line 114 – natural connection with which other parts of the cave? Perhaps indicate the underwater parts on the cave survey.

Changed to: The natural connection between the Réseau Sud and other parts of the Han-sur-Lesse Cave is fully submerged, but in 1960 an artificial tunnel was established facilitating access (Timperman, 1989).

965 Line 113-114. Rephrase. The sample can't have been collected in the Réseau Sud OR the southern network of Han-sur-Lesse
Changed to: The Han-9 stalagmite was collected within the Réseau Renversé, which is the most distal part of the Réseau Sud (Fig. 1B).

970 Line 115 – mark where the tunnel is on Fig. 1B.
It would be helpful to mark the position of Réseau Rénversé, Réseau Sud, Gouffre de Belveaux, Trou de Han on Fig. 1B
Figure 1B was revised. Locations mentioned in text were added, some information on original figure that was not relevant was removed.

975

The information about where the cave floods is confusing. Does flooding affect the sampling site?
Line 120 – when did the temperature logging take place? What were the sampling intervals?
Changed to: Temperature logging for six months with an interval of two hours in the Réseau Renversé shows an average cave temperature of 9.45°C with a standard deviation < 0.02°C, which

980 reflects the average temperature of 9.2°C above the cave for 2013.

Line 121 – why does the cave temperature reflect the average temperature for 2013, rather than the 1999-2013 average? Does this have implications for the interpretation later on?
Reply: Temperature in the cave is stable and reflects the average year temperature above the cave.

985

The important point here is that the sample comes from the part of the cave with the stable temperature. Was that likely the same in the past? Or is there evidence for other entrances in the past that may have affected ventilation in this part of the cave?
Changed to: For the Réseau Renverse, there are no indications that cave morphology changed

990 significantly since the last interglacial.

Line 137 – normally the half-lives are given as part of the U-Th data table. If you wish to give them in the text, why only give $_{230}$Th and $_{234}$U, and not the half-lives for the other relevant isotopes?
Reply: Half-lives were removed from text

995

Line 138 – specify atomic or activity ratio
Changed to: Ages were corrected assuming an initial 230Th/232Th atomic ratio of 4.4 ±2.2 x 10-6.

Line 139 – And also Shen et al., Geochimica et Cosmochimica Acta 99 (2012) 71–86
1000 Reply: Reference was added

What is the age datum?
Added: The age datum is 1950 CE

1005 Where were the stable isotopes measured?
Added: All stable isotope analysis were carried out at the Stable Isotope Laboratory, Vrije Universiteit Brussel.

Line 144 - 300μm diameter, radius, length?
1010 Changed to: For all samples, tungsten carbide dental drill bits with a diameter of 300μm from Komet were used.

Line 145 – Revise sentence beginning "In function of the growth rate...". It is unclear what is meant.
Changed to: As a function of the growth rate, 1000, 500 and 250μm sampling resolutions were
1015 applied in order to maintain a more or less equal resolution in the time domain.

Line 146 - why were samples kept at 50°C
Changed to: Samples were kept at 50°C prior to analysis to avoid contamination

1020    Line 149 – what is the in-house standard made from?
Changed to:  Two samples of the in-house standard MAR-2(2), made from Marbella limestone and which has been calibrated against the international standard NBS-19 (Friedman et al., 1982), were measured every 10 samples to correct for instrumental drift.

1025    Line 153 - do you mean replicate sample?
Changed to: At regular intervals, a replicate sample was measured in a different batch to check for the reproducibility of the analytical method.

Line 163 – it looks like the detrital material is on the base and not just the sides.
1030    Changed to: In the lower 15mm, some fine, brown detrital laminae can be seen, although they are confined to the very base and the lateral sides of the stalagmite.

Line 163 onwards - it would help to be more specific about the sections of the speleothem being discussed. E.g. layering is visible between 365-700 mm dft.
Reply: The entire section 4.1. was revised to make it more clear.

How are the discontinuities identified? Macroscopically? By ages?
Changed to: This discontinuity was identified macroscopically.

Line 174 - …dashed lines in figure 2D….
Line 174 – do you mean no macroscopically visible internal layering is present?
Changed to: Besides some subtle unconformities marked by the dashed lines in Fig. 2D, no internal layering is visible macroscopically.

Lines 178-181 – You say that the latter fabric (i.e. the coarser one) has smaller columnar calcite crystals, thus you describe it as "columnar open". You then contradict yourself and say that the "coarser morphology has substantially larger crystals…..defined as columnar elongated".
Changed to: Variations in fabric occur between macroscopically defined 'denser' and 'coarser' calcite (Fig. 2E), where the latter has smaller columnar calcite crystals with significantly more inter-crystalline porosity often filled with fluid inclusions (Fig. 3B), and thereby described as columnar open. The denser morphology has substantially larger crystals with almost no pore space and can be defined as columnar elongated.

Line 183 – How can the smaller more equant calcite crystals also cover the fine layer of brown detrital material that is D2.
Changed to: There, the columnar fabric is replaced with smaller more equant calcite crystals (Fig. 3C), which are then followed by a fine layer of brown detrital material representing D2.

Later in section 5 there should be some discussion about why the morphology changes throughout the stalagmite. Why does the fabric change from dense to coarse? Why is there visible layering in some parts and not others. What might cause the growth axis to shift for 20mm?

Line 187 – The datum is given here as 2015 CE. Yet in the table it is 1950 AD.
Include some text in section 4.2 about the concentration and cleanliness of the samples.
Changed to: In all samples, the detrital Th content, estimated by 232Th concentration and the initial 230Th/232Th atomic ratio, is relatively low (range 6419 – 208 ppt). This leads to only minor corrections for the 230Th age (Table 1).

Line 197 – be more specific about the section of stalagmite
Do you have any idea why there are age inversions?
Changed to: Between 176 and 0mm dft, the distribution of the ages is more chaotic, with the occurrence of several age inversions and outliers.

Line 198 - What are the distinctive changes in morphology?
Lower amplitude variability in both δ13C and δ18O occur in the lower part of the stalagmite, and larger amplitude variations are present from  ~400mm dft upwards, corresponding to distinct transitions in morphology (alternating zones of dense, more brown and coarser, more white calcite, Fig. 2E).

Line 199 – I think there should be some mention that the best test for isotopic equilibrium is to have a reproducible record (see Dorale and Liu, 2009). However, in the absence of a second speleothem, the Hendy test has been used instead. Do any of the modern monitoring studies record calcite deposition in isotopic equilibrium with the drip water?

Age model – why does the oldest part of the age model (figure 5) appear to miss out sample DAT-1? It is within stratigraphic order, yet the age model appears to completely miss it.

Reply: The entire part about Hendy tests was removed, as it is clear from review #2 that Hendy tests were not the best way to check for equilibrium deposition. In the discussion of the stable isotopes, a new section was added were covariation of stable isotopes is examined with Pearson's correlation coefficient. These coefficient were added in the new Table 2.

Added:

5.2.1 Isotopes deposited in isotopic equilibrium?

The best test for the presence of kinetic fractionation is to have a reproducible record (Dorale and Liu, 2009). However, in the absence of a second stalagmite record, Hendy tests could be performed (Hendy, 1971). The problem here is that growth rates are rather low and the layering very fine, so it would be hard to sample precisely in one layer. Therefore, an additional test for correlation of $\delta 13C$ and $\delta 18O$ was done by calculating the Pearson's correlation coefficient on the entire record and on the three growth phases separately (Table 2). Yet, a correlation between $\delta 13C$ and $\delta 18O$ does not give conclusive evidence for the presence of kinetic fractionation, as both $\delta 13C$ and $\delta 18O$ are expected to be controlled by climate and could therefore show positive or negative covariation (Dorale and Liu, 2009). The Pearson's coefficients reveal that there is a clear difference between the separate growth phases. The first growth phase, with $\rho = 0.024$, marks no covariation. The second growth phase, with $\rho = -0.467$, has a substantial degree of negative covariation, whereas the third growth phase ($\rho = 0.461$) has a positive covariation. The differences between the coefficients of the separate growth phases indicate that several processes are controlling the stable isotope variability and that the presence or absence of covariation reflects changes in climate conditions between the growth phases rather than the presence or absence of equilibrium. Nevertheless, equilibrium deposition between the drip water and recent calcite in Han-sur-Lesse Cave has been observed by Van Rampelbergh et al. (2014).

Lines 218-223 – Why does the age model need to be adjusted at the end of growth phase 2? Looking at the ages in table 1, DAT 6, 16 and 19 are all in stratigraphic order within the limits of dating uncertainty. Why then is there a discussion about whether DAT-19 is an outlier or not? Why has linear interpolation been applied at the end of growth phase 2? For this discussion, the names of the relevant samples should be added to figure 5.

Incidentally, it would be helpful to mark on one of the figures, either figure 2 or figure 5, what is meant by growth phases 1,2 and 3.

Reply: Figure 5 has been revised (now figure 4). Also, the entire discussion about the age model has been modified taking into account suggestions from review #1 and #2.

Changed to:

The StalAge algorithm (Scholz and Hoffmann, 2011) was applied to the individual ages in order to construct an age-depth model, displayed in Fig. 4. It is clear that the stalagmite endured three separate growth phases, and that the discontinuities, expressed in the stalagmite morphology at 302 and 176mm dft (Fig. 2D), correspond to two hiatuses separating these three growth phases. In the first growth phase, all ages are in stratigraphic order and are included within the model and the 2σ error. DAT-1 has only limited weight in the final model and an explanation for this is given in the algorithm specifications (Scholz and Hoffmann, 2011). During the modeling process, the StalAge algorithm has a step where the data is screened for the occurrence of minor outliers and age inversions. This is done by fitting error weighted straight lines through subsets of three adjacent data points. However, DAT-1 is located in the basal part of the stalagmite, so less subsets of three data points can be used including DAT-1. If DAT-1 does not fit on the error weighted straight line created with the adjacent data points DAT-10 and DAT-11, which is the case here, the error of DAT-1 will be increased and the weight of DAT-1 in the Monte Carlo simulation for the age fitting will decrease. This results in less solutions where DAT-1 is included in the Monte Carlo simulated age models. The occurrence of substantial changes in growth rate in the boundary areas of a speleothem sample is

recognized as a limitation of the StalAge algorithm (Scholz and Hoffmann, 2011). Even though the three growth phases were modeled separately with StalAge, the model does not perform well with the start of the Hiatus 2, as DAT-19 is completely excluded. Likely, this is again caused by the fact that DAT-19 is located in a boundary area. Here, the stalagmite petrography shows clear evidence of a significantly decreased growth rate after DAT-16 (110.6 ka), i.e. very dense, brownish calcite with fine laminae (Fig. 2C and E). In complex cases, such as in this study where multiple hiatuses occur, the simplest model is still the best. Therefore, linear interpolation combined with good observations of changes in petrography, was applied to include DAT-19 within the age model (Fig. 4, red line). For the third growth phase, because of the occurrence of several age inversions, the resulting age model is unreliable. Despite the fact that ages clearly cluster between ~103 and ~97 ka, the chronology of the third growth phase is only poorly constrained and therefore a detailed interpretation of Han-9 in terms of paleoclimate is limited to the first two growth phases. The first growth phase starts at 125.34 +0.78/-0.66 ka with stable growth-rate of 0.02mm yr-1 up to around 120.5 ka. After that, the growth rate significantly increases, with values up to 0.15mm yr-1. At 117.27 +0.69/-1.02 ka, growth ceases and the first hiatus, H1, starts. The hiatus lasts 4.41 +1.10/-1.49 ka and at 112.86 +0.47/-0.41 ka growth phase 2 starts. DAT-4 and DAT-5 were taken 6mm below and above the discontinuity and the age-depth model does not show any reason to question the timing of H1. As for the second growth phase, growth-rate remains at a constant pace of 0.04mm yr-1 until approximately 110.5 ka, where it decreases to 0.006mm yr-1. At 106.59 +0.21/-0.22 ka, the second growth phase ends. Given this age-depth model, stable isotopes were analyzed with a temporal resolution between 100 and 0.3 years, and an average of 16 years.

Lines 224-232 – Refer to figure 5 for the timing of growth phases. Where is the growth rate data plotted? What is the reason for the change in growth rate? What is the temporal range of the stable isotope sampling interval (i.e., not just the average).
Changed to: Given this age-depth model, stable isotopes were analyzed with a temporal resolution between 100 and 0.3 years, and an average of 16 years.

Line 235 – what is the reason for difference in d13C between modern and Han-9?
Lines 236-258 – This section on d13C needs some revision. The shift of 5‰ to enriched values in Han-9 is centred on c. 110 ka. How long does this isotopic excursion last for? Be more specific about when the grass assemblage at Eifel was at 40%. How reliable is the chronology on the maar lake record? After much discussion about the different factors that control d13C in speleothems, there is no decision as to which mechanism(s) are controlling d13C in this particular record. The end of section 5.2 concludes by saying that shifts in d13C are caused by changes in vegetation type. After assessing all the different causes of changes in d13C, why has this conclusion been reached? i.e., Why is prior calcite precipitation discarded? Furthermore, why is there a mismatch between the pollen record from Eifel and the d13C of Han-9 if changing vegetation is the control on d13C? Between c.126-114 ka broad leaf trees dominate the assemblage, hence one would expect a speleothem d13C of -6 to -14 ‰, and this is the case at c.-7‰. The enrichment in speleothem d13C is then in agreement with the decline in broad leaf trees and increase in grass assemblages. This is fine too. But then the speleothem d13C shifts to c. -9 ‰ after 109 ka, i.e. it is even more depleted than it was between 126-114 ka, yet at this time grasses are most abundant (60%) and trees much less (40%). If the shifts in speleothem d13C are really reflecting changes in vegetation, then the absolute d13C values and pollen assemblages from the maar lake are not in agreement.
Reply: Section about d13C has been revised

Line 260 – "precipitation" is a bit vague in terms of control on d18O. Perhaps use moisture source, amount effect.
Changed to: In mid-latitude Europe, several different processes (including temperature, amount effect and ocean source) influence speleothem δ18O variability (McDermott, 2004).

Lines 272-273 – "precipitation controls" is too vague. Do you mean amount? Is the North Atlantic Ocean the source for the precipitation all year round? And would it have been the source in the past?

Changed to: Temperature and precipitation (through the amount effect) controls are thus expected to contribute most to the speleothem δ18O variability

Added in section 2: The dominant moisture source in northwestern Europe is the North Atlantic Ocean, and this remains constant throughout the year (Gimeno, 2010).

Line 276 – global d18O values of what?

Changed to: Waelbroeck et al. (2002) estimated that during MIS 5d, average global δ18O values of ocean waters were up to 0.5‰ higher compared to MIS 5e.

Lines 283-288 – be more specific about which sections of the d13C signal are controlled by the different mechanisms.

Reply: Paragraph was removed.

Lines 296-297 – The results presented here show quite clearly that speleothem deposition is not restricted to optimum interglacial conditions.

Changed to: Cave systems in Belgium are known to be very sensitive recorders of glacial/interglacial changes, with speleothem deposition only during interglacial intervals (Quinif, 2006).

Lines 316-317 – Be specific about why this demonstrates that interglacial conditions were already present at 125.3 ka.

Changed to: The low δ13C values perhaps demonstrate that interglacial optimum conditions were already present before 125.3 ka, but that an increase in moisture availability caused by enhanced precipitation above the cave, shown by the δ18O decrease, was the factor needed to trigger growth of Han-9.

Line 321 – refer to figure at end of first sentence

Changed to: The isotope records of Han-9 are relatively stable between 125 and ~120 ka (Fig. 5).

Line 324-326 – revise sentence so that the time period that is being referred to is clear

Changed to: The long-term trend, as displayed by a fitted 7-point running average (Fig. 5), shows lower variability between 125 and ~120 ka, especially when compared to younger growth periods of Han-9 (i.e. 120-117.3 ka and 112.9-106.6 ka).

Line 329 – refer to figure 7 at end of list about speleothems and ice cores

Changed to: During 125-120 ka, other paleoclimate records display stable interglacial conditions, such as speleothems from the Alps (Meyer et al., 2008; Moseley et al., 2015) and from Italy (Drysdale et al., 2009), and other archives including ice cores (NEEM community, 2013) (Fig. 5)

Line 331 – reference for enhanced Greenland melting

In marine records off the Iberian Margin, the 125-119 ka period was identified as an interval of 'sustained European warmth', following a time of enhanced Greenland melting between 131.5 and 126.5 ka (Sánchez Goñi et al., 2012).

Line 332 – you say there is a constant growth rate, but there are only three data points through which the age model passes over nearly a 9 cm interval. I come back to the earlier comment that the age for DAT-1 is completely ignored by the age model, despite being in stratigraphic order. If the age model took this age into account, then the growth rate would not be constant over the interval 120-125 ka.

Reply: No growth rate changes since original age model is retained.

Line 333 – no significant change in what over time? What is it about the layered calcite that indicates a stable climate?

Changed to: This is also supported by the constant growth rate (Fig. 4) and the speleothem morphology, displaying a sequence of layered calcite which does not show any significant change over the 125-120 ka period (Fig. 2C-E).

Line 336 – refer to figure at end of first sentence.
Line 336 – don't use "onwards", be specific about the time interval being discussed. This is important for the sentence where it is stated that no major changes in d13C are observed. Major changes take place at 117.5 ka (as is discussed in the next paragraph), and this is "onwards" of 120 ka. Line 336-337 – This needs rewording. At 120 ka, there is indeed an enrichment in d18O of 0.5‰, and the growth rate does change at this point, but the reduction in diameter doesn't appear to happen until later within this growth period. Also refer to figure 2 regarding growth diameter.

Changed to: At 120 ka, an increase in δ18O of 0.5‰ is observed (Fig. 5). This change in δ18O of the speleothem corresponds with an elevated growth rate (Fig. 4) and a speleothem morphology that becomes progressively coarser, with layers that are less expressed (Fig. 2C and E).

Line 336-352 – In the results section there was a lot of text about the d18O being controlled predominantly by temperature, but in this section it has now switched to both amount and source effect. Also, why does it switch from amount to source?

Line 355 – the sentence beginning "As the increase…." isn't complete. It is only the first part of a thought. What is happening as a result of the increase in d13C?

Changed to: The increase in δ13C here is believed to reflect changes in vegetation, such as an increase grasses resulting in lower vegetation activity, linked to a changing (drying and/or cooling) climate.

Line 358-361 – yes the beginning of the hiatus takes place within GS 26, but the age 117.3 ka doesn't stand out as a particular event within Greenland. GS-26 began at 119.1 ka (Rasmussen et al., 2014)

Not changed: First of all, in the NGRIP δ18O record it falls within what is identified as Greenland Stadial 26 (NGRIP Members, 2004). Although the signature of this GS may not be as clear as the younger GS 25 or 24, it corresponds with the overall decreasing trend observed in the ice δ18O, and also recognized in the more recent NEEM ice core (NEEM community, 2013).

Line 370-371 – add the Meyer et al 2008 data to figure 7.
Reply: Figure was revised (now figure 5) and Meyer et al 2008 was added.

Hiatus 1 – boreal forests recover after the LEAP event, but the speleothem doesn't start growing until much later. Please comment.

Added in section 5.3.4: The LEAP event in the Eifel maar only lasts 468 years (Sirocko et al., 2005), yet speleothem growth does not recover immediately after the LEAP event. A similar observation was made for the start of speleothem growth at 125.3 ka: optimum conditions were already present before Han-9 started growing. From this delayed growth, it appears that climate conditions need to be more favorable (warmer/wetter) to initiate growth than to sustain growth.

Line 390 – use the GICC05modelext timescale rather than GICC05
Line389 – in Rasmussen et al 2014, GS 25 is 110.6 to 108.3 ka

Changed to: The maximum in δ13C and the minimum in δ18O correspond well with the timing of GS 25 (110.6-108.3 ka, Rasmussen et al., 2014) observed in the NGRIP record (plotted on the GICC05modelext timescale), implying that the stable isotopes of Han-9 reflect the temperature decrease of the stadial, which is likely since higher δ13C is linked to a less active vegetation cover during colder periods (i.e. more grasses) and lower δ18O is caused by lower temperatures.

Line 396 – there is a different timing here for GS25 compared to line 389
Line 396 – suggesting what is mainly temperature controlled?

Changed to: Between 112.9 and 111 ka, the variability of δ13C and δ18O in Han-9, predating GS 25, has an inverse relationship suggesting that δ18O is mainly temperature controlled.

1310 Line 400-401- Rephrase the question. You ask why speleothem growth continued during GS 25, but then answer why it stopped during GS24.
Changed to: However, if the hiatus has any affinity with GS 24, this raises the question why speleothem growth stopped during GS 24 and continued during GS 25. A plausible explanation could be that growth never fully recovered from the GS 25, and that less favorable conditions
1315 (cooler/dryer) during the GS 24 interval were sufficient to cease growth.

Line 404 – Growth rate decreased at the same time d18O increased (amount effect=less rainfall), but d13C decreased
Reply: Adressed in the section about stable isotopes
1320

Line 411-412- weak ending. Yes the age model is poorly constrained, but the ages clearly cluster within the interval 97-103.6 ka, hence, the more depleted d13C appears to be in good agreement with the high abundance of grasses. Perhaps it is a problem with the Eifel Maar chronology, rather than the U-Th chronology? You should discuss whether PCP or fractionation effects might be
1325 responsible.
Reply: Last part about growth phase three is removed, the depleted d13C (and PCP) is addressed in d13C section (5.2.2)

**Technical corrections**
1330 **General comments – There are many issues related to grammar, use of articles, prepositions, language, and in-house standards. Please address them.**
All suggested technical corrections in text were carried out.

1335

1340

**REVIEW #2**

1345

**General Comments**
Section 4.1 Speleothem morphology: This entire section is rather unclear.

The U/Th dates between 0 and 176 mm dft are unusable. I recommend not interpreting your record
1350 past Hiatus 2.

I lack confidence in the Hendy Tests (see Dorale and Lu, 2009). The first potential problem is that the drill bit is 300 microns and the reported growth rates are 20, 40 and 150 microns. This suggests that at worst the authors are averaging 15 years and at best, 2 years. Another potential explanation for
1355 the very flat Hendy Test results for d13C is that these tests only extend to 15 mm from the central growth axis which is likely within the splash cup and not enough distance (even under conditions of kinetic fractionation) for degassing to occur as the drip progresses toward the flanks. Also, the variability in the d18O data along a (presumably) single growth axis is nearly 25 % of the total d18O variability through time. This is less than inspiring. I suggest that a better test for kinetic fractionation
1360 is to look at how d18O and d13C covary (or anticorrelate) over the length of the record. Superficially I did this and it appears that there are intervals of strong anticorrelation such as during the 'Eemian

Optimum' and other periods of covariation such as after Hiatus II. Perhaps a running Pearson's could shed some light on when conditions favoured kinetic fractionation and this in itself provides valuable climate information. The presence of kinetic fractionation really only comes into play when aiming to apply the palaeotemperature equation. If this is not the goal (and I believe it is not here) then the presence or absence of kinetic fractionation does not rule out but can contribute information to the overall palaeoclimate interpretation.

Reply: In Section 4.3 (Results – Stable isotopes) the part about the Hendy test was removed, as well as Figure 4. This has been replaced by the Pearson's test in the discussion section. Within the discussion section, a new sub-division was made between equilibrium deposition – d13C – d18O.

For the Age Model, the authors might want to try COPRA to see if it handles the hiatuses better. Did the authors run StalAge over the two growth intervals separately? I don't see why StalAge would interpolate incorrectly over the last 2 dates in Growth Phase II if it were run over the two growth intervals separately. I suggest that short of trying COPRA, the authors should rerun StalAge over Growth Interval II alone and see if the age model is more true to the U/Th dates and errors over that interval. I think that the current linear interpolation is unsatisfactory. Again, I would limit discussion of Growth Phase III due to dating uncertainties.

Reply: Section 5.1 (Discussion – age model) has been completely revised and major issues, as addressed in the authors comment , were added. Also the figure has been revised.

Section 5.3 'A late onset of the Eemian' seems to be special pleading. Han-9 begins growing during the Eemian....in fact, well into it. I don't think there is enough evidence to support a late onset and would delete any of the discussion to this effect.

Changed to: 5.3 125.3 ka: Start of speleothem growth triggered by an increase in moisture availability

Discussion of controls on Han-9 d18O: 1) Lower T will cause lower d18O due to rainout, distillation, etc. 2) ice build-up will sequester 16O, leaving behind a higher ocean d18O (source moisture) and driving rainfall d18O higher, 3) lower T would cause lower in-cave T and drive stalagmite d18O higher (Craig equation) but this effect is trivial. Effects 1 and 2 act against each other but one would win out (unless they cancel each other out entirely!). It is likely that lower T and increased rainout/fractionation will win, and rainfall (and thus Han-9 d18O) will be more negative during colder conditions. It seems that the authors are arguing both sides in the manuscript but they need to pick one interpretation that applies to the whole record.

Added in section 5.3.4: The ice build-up effect, displayed by the increase in δ18O between 120 and 117.3 ka, is cancelled out by the effect of lower temperature, causing a decrease in speleothem δ18O

It's interesting and somewhat puzzling that Han-9 growth seems to be restricted to intervals between Insolation max and mins but not during (i.e., no growth centred on 104, 115, and 127 kyr). It would be interesting to know if there is some explanation for this
.
The authors use 'ka' and 'kyr BP' in the manuscript. Select one and be consistent throughout.

Reply: Changed throughout

I find the match between Han-9 d13C and the pollen record particularly compelling at least until Growth Phase III when dating (and possibly the data) inaccuracies dominate.

I disagree with stating that the start of Hiatus I marks the 'end of the Eemian', (such as the authors have done in the Conclusions and possibly elsewhere), rather, the authors could state that Han-9 ceases growth (Hiatus I) at 117.3 kyr, before the end of the Eemian as recorded by other proxies (references) suggesting that a critical threshold was reached in which conditions no longer favoured Han-9 deposition. This was possibly linked to a change in vegetation dynamics.

Reply: Not changed. 117.3 ka is identified as the end of the Eemian in this study. Also, it corresponds to what other speleothem studies recognized as end of the Eemian (Meyer et al., 2008; Hölzkamper et al., 2004)

**Enumerated Comments**

All suggested technical corrections in text were carried out.

**Figures**

**Figure 1**: Location of core MD03-2664 was added in Fig. 1A. In Fig. 1B, the terms in the text were better displayed.

**Figure 3**: Figure 2d shows 6 thin sections. Which ones are shown here in A-C. Suggest labelling Fig. 2d more clearly so one can more easily cross reference Figs. 2d and 3 A-C. In Fig. 2 the blue boxes and thin section labels are shown more clearly now. Locations of Hendy tests were taken out because these are left out in the text as well.

**Figure 4:** Hendy tests were left out.

**Figure 5**: The section after Hiatus 2 is almost unusable. The error envelope shown (grey shading) does not cover the entire interval of age inversions and increased dating error. This needs to be expanded to cover the whole interval. I recommend excluding this growth interval from your discussions/interpretations. There is nothing obvious in Table 1 to explain why the dating is so imprecise over this interval.

Also, in Figure 5, it would help if the authors would label the datapoints. The Age model discussion in section 5.1 refers to DAT-#'s quite a bit and it would help if the reader could make quick reference to a well-labelled Fig. 5.

Label Fig. 5 growth intervals and growth rates

Suggestions were taken into account. Fig. 5 is now Fig. 4.

**Figure 6:** This figure is redundant considering that the same data are provided in Figure 7. I suggest deleting. In Figure 7 just darken slightly the non-averaged d18O and d13C curves…they are a bit too light to see right now.

Figure 6 was removed.

**Figure 7:** This figure is not very well laid out and could be vastly improved. The Han-9 data is minimised with compressed axes while the previously published datasets are at the forefront. Consider compressing the published data and expanding the axes on the Han-9 isotope data. I suggest making all of the axes the same length.

It might be informative to move the Insolation record down over (or between) the Han-9 datasets.

The U-Series datapoints should be plotted on 2 lines only (or one when they are not overlapping). The way they are now, suggests that there is some monotonic increase through time. Misleading.

I suggest adding in the Yuan et al 2004 Dongge Cave data. I know this is a SE Asian Monsoon dataset but it tracks Greenland ice core data quite well and could help fill in some blanks.

Add interpretations to figure. For instance, along axes like Han-9 d13C add down arrows (towards heavier d13C values) labelled with 'increasing C4 plants', 'Colder/drier', and 'increasing PCP', 'decreasing soil bioproductivity'. For Han-9 d18O add down arrows (towards lighter d18O values) decreasing ice volume

Make certain that all axes are pointing in the correct direction regarding a uniform interpretation for the figure.

The authors highlight the hiatuses but not the actual climate info. I suggest just labelling the data breaks as 'Hiatus 1' and 'Hiatus 2' (if the authors choose to keep growth phase III), but using shaded rectangles and more obvious labelling to highlight the climatic episodes discussed in the text (i.e., the Eemian climatic optimum from 120 to 125). Perhaps helpful to the reader would be to have the climatic periods identified (for instance in Paragraph 1 of the Introduction) in horizontal shaded rectangles at the bottom of the figure below the pollen data.

In caption add d18O to 'NALPS speleothem record'. Add (SST) after 'Sea Surface Temperature' in B) if this is the first mention. Otherwise, just use SST.

I suggest adding in the Irvali et al 2012 Fig. 3 b MD03-2665 Planktonic d18O record to Fig. 7, possibly overlapping with the Sanchez and Goni SST record. There is a lot more detail in the Irvali record and it is discussed in the text.

Figure completely revised with suggestions from both reviewers.

**Response to additional Editor comments:**

Line 66-67: This sentence still makes no sense to me. What does it mean for D/O cycles to have "an affinity" to something? Please clarify.

Reply: The part "and have affinity to the cold events in the North Atlantic Ocean" was removed. The exact relation of the surface cooling and Greenland stadials is still debated, as mentioned further in the text.

You have stated in the methods that the 230Th/232Th ratio has been corrected with a certain value. Please provide a brief justification for the value chosen in the text.

Moved to results section and changed to: Atomic $^{230}$Th/$^{232}$Th ranges between 1064 and 33652 × 10$^{-6}$. Because detrital Th is relatively low in all samples, ages were corrected assuming an initial bulk earth $^{230}$Th/$^{232}$Th atomic ratio of 4.4 ±2.2 × 10$^{-6}$.

You have cited 3 papers for methods, yet some of these papers differ in their methods used. Can you please clarify which parts of the method come from which papers, e.g., the chemical separation of U and Th is described in XX whereas the instrumental method follows the procedure described in XX, etc.

Changed to: The applied method is based on the fundamental principles of U-Th dating and mass spectrometry on carbonates provided by Edwards et al. (1987). For state of the art improvements on sample preparation, MC-ICP-MS protocols and $^{230}$Th and $^{234}$U half-live values, see Shen et al. (2012) and Chen et al. (2013) and references therein.